# Magmas near the critical degassing pressure drive volcanic unrest towards a critical state

Giovanni Chiodini[1], Antonio Paonita[2], Alessandro Aiuppa[2,3], Antonio Costa[1], Stefano Caliro[4], Prospero De Martino[4], Valerio Acocella[5] & Jean Vandemeulebrouck[6,7]

During the reawaking of a volcano, magmas migrating through the shallow crust have to pass through hydrothermal fluids and rocks. The resulting magma–hydrothermal interactions are still poorly understood, which impairs the ability to interpret volcano monitoring signals and perform hazard assessments. Here we use the results of physical and volatile saturation models to demonstrate that magmatic volatiles released by decompressing magmas at a critical degassing pressure (CDP) can drive volcanic unrest towards a critical state. We show that, at the CDP, the abrupt and voluminous release of $H_2O$-rich magmatic gases can heat hydrothermal fluids and rocks, triggering an accelerating deformation that can ultimately culminate in rock failure and eruption. We propose that magma could be approaching the CDP at Campi Flegrei, a volcano in the metropolitan area of Naples, one of the most densely inhabited areas in the world, and where accelerating deformation and heating are currently being observed.

[1] Istituto Nazionale di Geofisica e Vulcanologia, sezione di Bologna, via D. Creti 12, 40128 Bologna, Italy. [2] Istituto Nazionale di Geofisica e Vulcanologia, sezione di Palermo, via U. La Malfa 153, 90146 Palermo, Italy. [3] DiSTeM, Università di Palermo, via Archirafi 36, 90123 Palermo, Italy. [4] Istituto Nazionale di Geofisica e Vulcanologia, sezione di Napoli Osservatorio Vesuviano, via Diocleziano 328, 80124 Napoli, Italy. [5] Dipartimento di Scienze, Università Roma Tre, Largo San Leonardo Murialdo, 1, 00146 Roma, Italy. [6] Université de Savoie Mont Blanc, Chambéry 73000, France. [7] CNRS, ISTerre, F-73376 Le Bourget du Lac, France. Correspondence and requests for materials should be addressed to G.C. (email: giovanni.chiodini@ingv.it).

Volcanic eruptions[1,2] are the surface manifestations of the final stages of crustal emplacement of mantle-sourced magmas. Understanding the transition of a volcano from quiescence to eruption is relatively straightforward at the frequently active mafic volcanoes, where the rates of magma ascent and the separation of magmatic volatiles drive pressurization of the magmatic systems and finally eruption[3–7]. In contrast, interpreting volcanic unrest is difficult at silicic volcanoes, since they commonly develop pervasive hydrothermal systems during their long repose periods[8,9]. The complex magma–hydrothermal interactions that result as magma finally makes its way to the surface during the reawaking of a volcano will modulate the physical and chemical signals recorded at the surface[10–14], and determine whether the magma will ultimately erupt[15].

Such magma–hydrothermal interactions are particularly complex and unpredictable at active calderas, where the hydrothermal circulation is particularly intense at the subsurface due to major structural control[16,17]. This is especially true for Campi Flegrei caldera (CFc), a long-lived resurgent caldera in the metropolitan area of Naples that was formed by the 39-ka Campanian Ignimbrite supereruption, which was the largest in Europe during the past 200 ka (ref. 18). Since the 1950s, CFc has been showing clear signs of potential reawaking, as indicated by frequent episodes of ground uplift (with a total of > 3 m of permanent cumulative inflation at the caldera centre[19]), shallow seismicity[20], and a visible increase in hydrothermal degassing[14]. After a period of major unrest in 1983–1984 characterized by thousands of earthquakes and a rapid uplift (~1.8 m over 2 years[19]), CFc subsided until 2005, when a new inflation started, resulting in a minor (~0.4 m over 10 years) but temporally accelerating uplift. The involvement of magma as a causal factor of the current CFc unrest is strongly supported by the composition of volcanic gas[21] and deformation changes[22]. However, it is not clear whether this unrest will culminate in an eruption and, if it does, over what timescale this will occur. The presence of more than half a million people living in the proximity of the caldera makes this situation particularly challenging for local authorities and other decision-makers, and highlights the urgency of obtaining a better understanding of interactions between the magma driving the unrest and its overlying hydrothermal system.

While it is universally accepted that the injection of new magma is a common mechanism that drives hydrothermal systems towards the critical state[23,24], the mechanisms and timescales of magma–hydrothermal interactions during unrest remain poorly understood and difficult to forecast[16]. One key aspect that has received little attention is the role that magmatic gases may play in heating the hydrothermal system, and ultimately in driving the unrest.

The present study linked magma degassing at depth with the resulting perturbation in the overlying hydrothermal system. Here we initially use the results of volatile saturation[25] models to demonstrate that decompressing magmas can reach a critical condition, which we refer as a critical degassing pressure (CDP), at which their ability to release water and convectively transport heat is increased by a least an order of magnitude. We then use physical models[26] to show that magmatic volatiles released at the CDP, when injected into an overlying hydrothermal system, lead to extensive heating and expansion, and cause temporally accelerating ground deformation. Finally, we examine ground deformation time series from CFc and some other restless calderas, which identifies consistent accelerating ground uplift trends that are reminiscent of those predicted by our model. We conclude that magmas at the CDP can be recurrent causal factors in driving volcanic unrest towards a critical state; that is, a state

near a bifurcation at which the evolution of the system can either culminate in an eruption or change trend and cool down[27].

## Results

### The critical degassing pressure during magma decompression.
The decompression of fresh magma results in the selective release of dissolved volatiles depending on their solubilities[28]. This means that while barely soluble $CO_2$ dominates deep degassing[3,29,30], more-soluble $H_2O$ prevails at shallower depths[31]. Given this selective release of volatiles from magma and the different capacities of these two species to carry thermal energy, the pattern of heat transfer to overlying rocks and hydrothermal systems will be complex and will vary as the unrest progresses.

Several saturation models have been reported in the literature[25,28,32–36] that can be used to investigate magma degassing in an $H_2O$–$CO_2$ system. Whereas most of them predict the $H_2O$ and $CO_2$ solubilities over narrow ranges of silicate melt composition, one of the models[25] can be extended to any silicate melt composition. Here we use that model to predict $H_2O$–$CO_2$ partitioning during magma decompression; however, the use of alternative models would lead to similar results and conclusions.

Model calculations were initialized at conditions relevant to CFc, where the marked variations in the compositions of fumarole emissions[37] observed over the last 30 years have been attributed[14,21] to decompression-driven open-system degassing of trachybasalt magma. A good match[21] between model-calculated solubilities[25] and experimentally derived solubilities for CFc or CFc-like magmas has also been demonstrated.

Figure 1a shows the total quantities of $H_2O$ and $CO_2$ released at equilibrium conditions by 1 kg of magma during open-system depressurization. We assumed isothermal conditions (temperature = 1,425 K) and that the $CO_2$–$H_2O$ mixture is initially saturated at 200 MPa. This pressure corresponds to a depth of ~8 km, where a large magma reservoir below CFc has been detected by seismic tomography[38].

A range of total volatile contents was explored, but all of the simulations converged to indicate different degassing behaviours of $CO_2$ and $H_2O$. Figure 1a shows that, during deep (pressure > 150 MPa) degassing stages, less than 0.001 moles per kilogram of magma of $CO_2$-dominated magmatic gas are separated for each 1-MPa decrease in pressure. $H_2O$ degassing becomes effective only at lower pressures (Fig. 1a,c), and when this happens there is a narrow pressure interval over which the total amount of separated fluid increases steeply (by more than one order of magnitude). This stage is marked by abrupt variations (Fig. 1a) and it leads to the complete exhaustion of $CO_2$ in the magma, leaving only $H_2O$ available for subsequent low-pressure degassing (Fig. 1c). A particularly important feature at this stage is that each model curve shows an inflection point (Fig. 1a) at a specific pressure, which we refer to as the CDP. In each of the model curves of Figs 1 and 2, we set the CDP as the pressure value at which the second derivative of separated gas content with respect to pressure reaches its maximum. This condition marks an abrupt increase in the amount of thermal energy released through fluid expulsion, from < 50 J to > 1,000 J per kilogram of magma and for each 1-MPa decrease in pressure (Fig. 1b).

We found similar nonlinear degassing behaviour over a range of magma compositions and volatile contents. For any magma type (for example, rhyolitic magma; Fig. 2a), complete $CO_2$ exhaustion in the melt marks a critical condition at which the amount of separated fluids and the energy transfer to the hosting rocks both increase dramatically. It is important to stress that most of the $H_2O$ (~95% of the original content) is still dissolved in the magma at the CDP (Fig. 1c) and is therefore available for

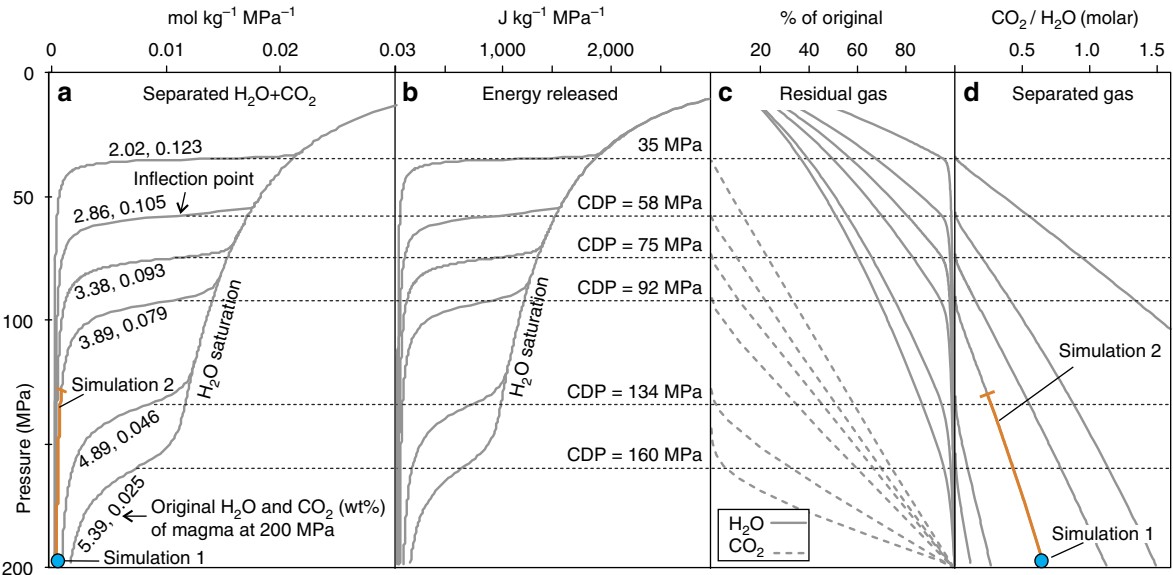

**Figure 1 | Open-system magma-degassing models for CFc.** The different curves refer to different initial $H_2O$ and $CO_2$ contents and describe the evolution of the fluids released during magma depressurization in an open-system Rayleigh-type degassing process (where at each infinitesimal decompression step, an infinitesimal parcel of gas phase in excess of the permissible saturation is distilled from the well-mixed magma). Theoretical degassing curves were calculated for the most-primitive magma compositions of CFc (trachybasalt) and show the pressure dependence of (**a**) the moles of $H_2O$ and $CO_2$ released by one kilogram of magma for each 1-MPa decrease in pressure; (**b**) the energy associated with fluid release (enthalpy of the separated $H_2O$–$CO_2$ mixture at the specific temperature and pressure, computed using MUFITS[63]); (**c**) the residual gas in the melt, as a percentage of the original content; and (**d**) the $CO_2$/$H_2O$ ratio of the released fluids. The blue circle and orange line indicate the conditions used in the corresponding TOUGH2 (ref. 26) simulations.

further subterranean gas–magma separation, heating, and (eventually) sustaining an eruption[39].

Our calculations were performed under open-system conditions since the long-lasting variations in the fumarole-gas composition at CFc cannot be reproduced in a closed system, instead requiring efficient separation of gas from the magma[21]. The large amount of magmatic fluids released by CFc manifestations[40] also supports an ongoing open (rather than closed) magma-degassing behaviour. Our open-system Rayleigh-type degassing model assumes that volatiles are continuously separated from magma at each decompression step. However, the release of magmatic fluid from CFc surface manifestations actually shows a pulsed (noncontinuous) behaviour (Supplementary Figs 1 and 2), which suggests a mechanism in which periods of closed-system decompression alternate with episodes of system opening and gas release (that is, a multistep degassing process) over timescales of years.

Tests show that such multistep degassing can be adequately reproduced as an open-system degassing process provided that there are numerous and recurrent system-opening events, as is likely to be the case.

Open-system, unsteady degassing is not only observed at CFc[41,42]. During extrusive volcanic eruptions, pulsed degassing behaviour can occur even at shorter timescales, and is thought to derive from multiphase flow dynamics within the conduit[43]. At CFc, we argue that such degassing behaviour can ultimately result from the complex geometry of crustal volcano plumbing systems, whose intricate networks of fractures, dikes, sills and small reservoirs[44–47] facilitate the segregation of gas from melt, and the loss of volatiles from a foam layer[48]. Foam growth in low-viscosity mafic melts takes place over timescales of months to a few years[48,49], which is faster than the observed decennial trends in gas composition. While we therefore favour an open-system scenario, we also show examples of model degassing simulations in closed-system conditions (Fig. 2b) to demonstrate that a CDP can be reached even in that type of system, despite the

mass of released volatiles varying less markedly than in open-system conditions. We conclude that the concept of the CDP applies over a wide range of magmatic conditions. We also find that CDP conditions are reached independently on the solubility model used, for example, VolatileCalc[36] (Supplementary Fig. 3).

The thermal regime of hydrothermal systems can be strongly impacted if the underlying magma approaches the CDP. Within the typical temperature and pressure hydrothermal range, $CO_2$ behaves as an incondensable species, while $H_2O$ can condense and therefore heat the rocks very efficiently. Our models of fluid flow in porous media[26] that describe the injection of fluids enriched in either $CO_2$ or $H_2O$ into a virtual hydrothermal system confirm the different heating capacities of the two volatiles. The ability of ascending magmas to heat any overlying hydrothermal system will therefore be greatly enhanced as the CDP is approached. We use below the CFc example to further illustrate this aspect.

**The case of Campi Flegrei caldera.** Of the several quiescent calderas worldwide, CFc has recently shown among the clearest signs of unrest. At CFc, several ktons of hydrothermal fluids are emitted daily by the Solfatara-Pisciarelli fumarolic field[40] (Fig. 3a,b). Stable isotopes of fumarolic steam concur to indicate that such fluids are, at least partially, sourced by magma degassing[50].

The large variations in the fumarole emissions of $N_2$–He–$CO_2$–Ar (ref. 21), including the 25-year-long decreasing trend of the $N_2$/He fumarole ratio (Fig. 3c), fully support the idea that a primitive magma degassing in open-system conditions at increasingly lower pressures is sustaining the unrest. A particularly important observation is that the ground deflation and $N_2$/He gas ratios followed exponential-like trends from 1985 to 2005, with very similar characteristic times, implying common source processes[14]. The presence of magma depressurization is also supported by modelling of the ground uplift in 2012–2013, which has been interpreted as the effect of a magma intrusion at a

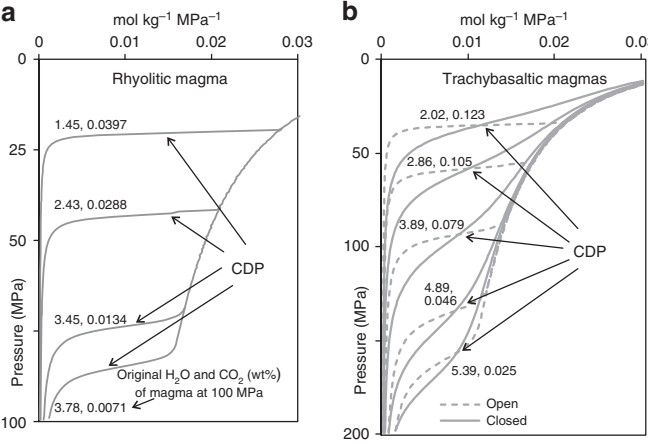

**Figure 2 | Results of magma degassing models.** The different curves refer to different initial $H_2O$ and $CO_2$ contents, reported as couple of values on each and describe the evolution of the fluids released during magma depressurization in various conditions. (**a**) Theoretical degassing curves were calculated for rhyolitic magma compositions and show the pressure dependence of the moles of $H_2O$ and $CO_2$ released during open-system degassing by one kilogram of magma for each 1-MPa pressure decrease. (**b**) Theoretical degassing curves were calculated for the most-primitive magma compositions of CFc and refer to open and closed degassing (trachybasalt).

depth of 3 km (ref. 22). At the same time, a generalized heating up of the CFc hydrothermal system is indicated by the 15-year-long exponential increase in CO emissions from the fumaroles (Fig. 3d); note that CO is the fumarole gas most sensitive to temperature changes[51].

Based on these observations, we argue that the CFc magmatic system may be approaching the CDP; that is, that depressurizing magma may release fluids progressively richer in $H_2O$ so as to affect the thermal structure of the hydrothermal system. We tested this hypothesis by using TOUGH2 (ref. 26; see Methods) to model the injection of magmatic fluids (IMF) into a hydrothermal system under physical conditions appropriate for CFc[13] (Fig. 4).

Our new model simulations refine previous ones[13] that first identified the magmatic gas trigger of the unrest. The model involves injecting $H_2O$–$CO_2$ magmatic gas mixtures into a virtual hydrothermal system at subcritical temperature and pressure conditions. The composition ($CO_2/H_2O$ ratio) of the injected magmatic gas phase is based on the results of our magma-degassing models (see Methods and Fig. 1d). We highlight that these modelled magmatic $CO_2/H_2O$ ratios can only approximate the composition of fluids entering the real hydrothermal system, since the model does not account for secondary processes potentially occurring along the magma-to-hydrothermal gas cooling path.

We simulated 14 IMF events occurring between 1983 and 2014, whose timing and intensity were constrained based on

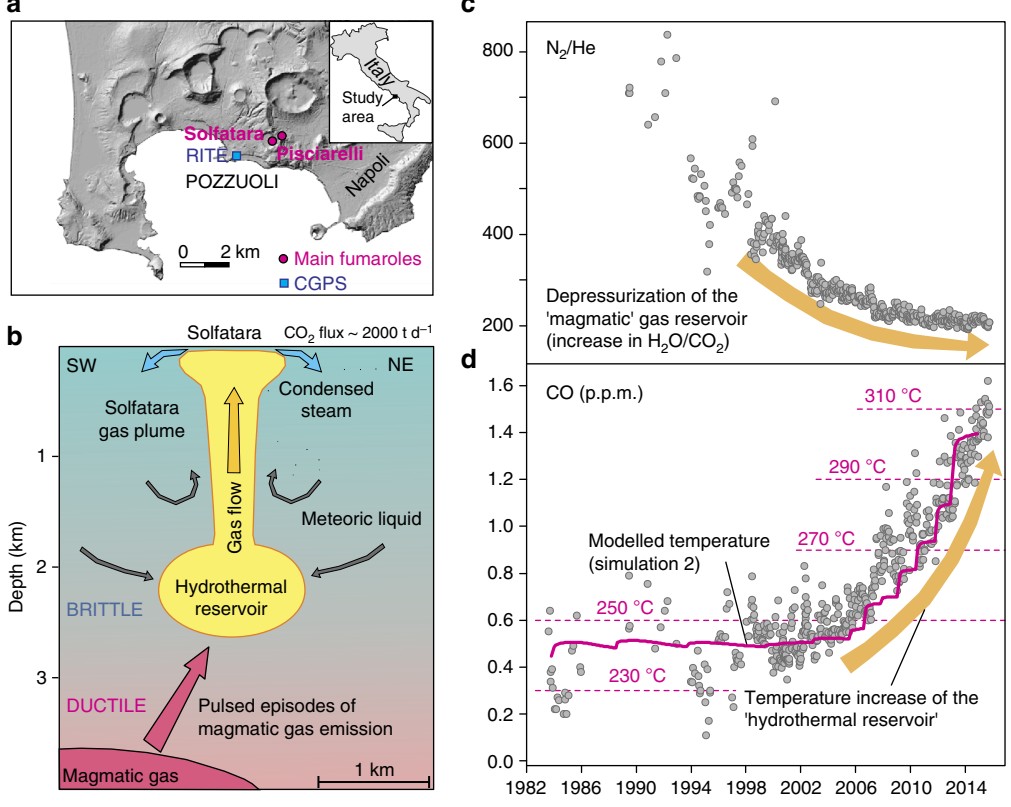

**Figure 3 | The hydrothermal system of CFc and its signals.** (**a**) Locations of CFc and the main hydrothermal sites: Solfatara and Pisciarelli. (**b**) Conceptual model of the hydrothermal system feeding the two manifestations: a 4-km-deep zone of magmatic gas accumulation that supplies fluids to a shallower part where they vaporize liquid of meteoric origin to form a 2-km-deep vertical plume of gas[14]. Previous geochemical investigations based on the stable isotopes of water revealed the presence of typical magmatic waters in the Solfatara fumarole vents[50]. (**c**) Temporal evolution of the $N_2/He$ ratio at the Solfatara fumaroles. (**d**) Time series of the CO content in the Solfatara fumaroles. The increasing trend indicates heating of the system, and matches the TOUGH2 (ref. 26) model-derived temperatures (magenta line) exceptionally well.

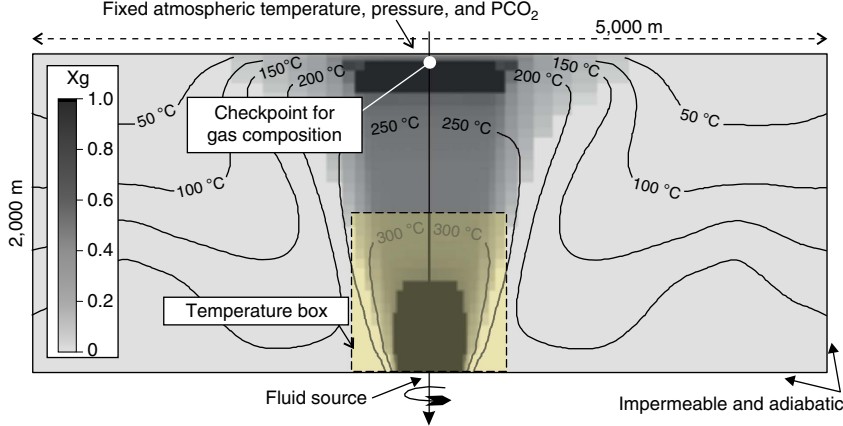

**Figure 4 | Computational domain of the TOUGH2 simulations.** The rock physical properties were homogeneous (porosity = 0.2; thermal capacity = 1,000 J kg$^{-1}$ °C$^{-1}$; density = 2,000 kg m$^{-3}$; horizontal permeability = 10$^{-14}$ m$^2$; vertical permeability = 1.5 × 10$^{-14}$ m$^2$; and thermal conductivity = 2.8 W m$^{-1}$ °C$^{-1}$). The temperature and the volumetric gas fraction Xg (different shades of gray) refer to steady-state conditions. The 'checkpoint for gas composition' is the zone where the simulated $CO_2/H_2O$ is compared with the measured one (see Methods). The 'Temperature box' (yellow rectangle above the injection zone) is the region where the average temperature is calculated during the simulations (see Figs 5 and 6 and Supplementary Fig. 1c).

measured geochemical anomalies only (see Methods and Supplementary Fig. 1a,b). Ground deformation pulses and clustered earthquakes support the timing of the IMF events, which were independently fixed based on geochemical anomalies (Supplementary Fig. 2). Previous simulations[13] considered injections of hot fluids with a constant $CO_2/H_2O$ ratio. Here we update these previous calculations to the current CFc state (Simulation 1), but also consider a new scenario (Simulation 2) in which magmatic fluids that are increasingly rich in $H_2O$ are injected. The first scenario corresponds to degassing of a stationary source at 200 MPa and with original $H_2O$ and $CO_2$ contents of 3.89 wt% and 0.079 wt%, respectively (blue circle in Fig. 1a,d). The second scenario describes the depressurization of the same source down to 130 MPa (orange line in Fig. 1a,d). The $CO_2/H_2O$ ratio used in each IMF event was inferred from the measured $N_2/He$ ratios and the results obtained in simulations of open-system magma-degassing models (Supplementary Figs 4 and 5).

We found that each modelled IMF episode involves the injection of 0.1–25 Mt of magmatic fluids, which is within the range of the gas mass associated with small to moderate-size volcanic eruptions[13]. The modelled cumulative trends of injected magmatic fluid masses exhibit clear exponential acceleration since the 2000s (Supplementary Fig. 1d). The acceleration trend is steeper in Simulation 2 (in which the gas compositions varied during the simulation) than in Simulation 1. Simulation 2 also predicts an average temperature increase of 60 °C in the deep-central part of the hydrothermal system (Figs 3d,5 and 6). One interesting outcome of Simulation 2 is that, while the $CO_2/H_2O$ ratio of the injected magmatic fluids decreases over time, the simulated gas composition at the 'checkpoint for gas composition' (Fig. 4) becomes increasingly rich in $CO_2$ (Supplementary Fig. 1b). This apparent paradox results from $H_2O$ condensation in the hydrothermal system, which is the same process heating the rocks. Condensation of a mixed magmatic-meteoric vapor, followed by $H_2O$–$CO_2$ oxygen isotope exchange in the fumaroles' feeding conduits[50], also well account for the observed hydrogen and oxygen isotope composition of fumarolic steam (Supplementary Fig. 6).

Our model predicts that the injection of increasingly $H_2O$-rich volatiles released by magma approaching the CDP leads to significant heating of the hydrothermal system (Fig. 5a)

with obvious implications for volume expansion of the rocks (Fig. 5c).

Independent observations support these model predictions of the thermal evolution of the CFc hydrothermal system. The modelled pattern of temperature increase (Fig. 5a) mimics the observed increasing trends in CO content of the fumaroles (Fig. 3d) and temperature estimations obtained from the CO–$CO_2$ geothermometer[14] (Fig. 5b). Concurrently with the heating onset predicted by the models, the Pisciarelli fumaroles located on the eastern slope of Solfatara visibly increased their flow rates and temperatures in 2005–2006, with recurrent episodes of mud emissions, the formation of new vents, and localized seismic activity[14] (Fig. 6). Furthermore, comparison of the locations of earthquakes in 1983–1984 and 2005–2014 highlights the recent disappearance of shallow seismicity in the Solfatara area, suggesting a transition from an elastic-like to a plastic-like behaviour, which is probably due to heating of the rocks[52].

## Discussion

We demonstrated, for the first time, a discontinuity in the degassing pathway of decompressing magmas. Attainment of this critical condition is purely controlled by the differential solubility of magmatic volatiles. Magmas approaching the CDP abruptly increase their $H_2O$ release efficiency, and as such enhance their ability to convey heat to the overlying hydrothermal system. Steam-heated host rocks surrounding magma may, in turn, alter their physical properties, lowering their mechanical resistance. Steam injection, which is a technique used in the oil industry for extracting heavy oil[53], demonstrates that heating can affect ground deformation to a much larger extent than that expected from pure thermal expansion. Steam injection and the associated latent heat release deform the media also via thermally induced shear dilation[54] and enhancing fluid-flow permeability. We conclude, therefore, that the CDP concept may contribute to explain acceleration in signals observed during volcanoes unrests.

We propose the CDP can help interpreting the current evolution of the CFc unrest. Escalation in hydrothermal activity, and increasing concentrations of fumarolic species more sensitive to temperature, point to an ongoing heating process of CFc. Our thermo-fluid-dynamic models here suggest that injection of magma-derived $H_2O$, becoming more voluminous and frequent

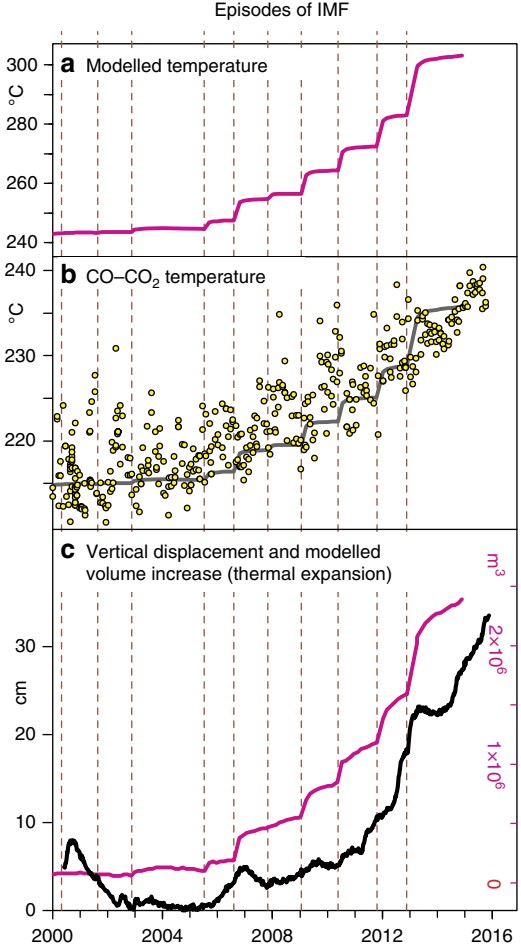

**Figure 5 | Observations and modelled data for the current period of unrest of CFc.** (**a**) Average temperature obtained by the model (in Simulation 2) for the central deeper zone of the computational domain. (**b**) Temperatures computed using the CO–CO$_2$ geothermometer at Solfatara fumaroles[14] compared with modelled temperatures (gray line). The modelled temperatures, which refer to the central deeper zone of the computational domain ('Temperature box' in Fig. 4), have the same temporal evolution but are systematically higher than the CO–CO$_2$ temperatures, which reflect the thermal state of the upper part of the hydrothermal system[14]. (**c**) Vertical displacements measured at the RITE CGPS station (black line) and modelled volume increases in the computational domain due to thermal expansion (magenta line). We used a coefficient of volumetric thermal expansion of $3 \times 10^{-5}\,°C^{-1}$.

in time, may well be controlling such heating. We caution that, since our model does not account for rheological properties and heterogeneities of the system, it cannot be used to predict mechanical evolution, for example, caldera deformation. We yet note a similarity between the temporal evolution of caldera uplift and our modelled hydrothermal temperature increase (Fig. 5c). We conclude, therefore, that magmas approaching the CDP may be contributing to accelerating caldera inflation, observed since 2005. At CFc, thermal effects can strongly be enhanced by specific physical characteristics of the rock matrix (Neapolitan Yellow Tuff), which is particularly sensitive to thermal alterations due to the presence of thermally unstable zeolites[55] (for example, the tensile strength of these rocks can halve when the temperature increases from 100 to 300 °C (ref. 56)).

CFc is unlikely to be an isolated case. A similar, several-year-long period of accelerating inflation possibly driven by magma depressurization and heating processes was observed before

eruptions at Rabaul, Papua New Guinea[16] and Sierra Negra, Galapagos[57]. Like at CFc, the accelerating deformations at these systems can be empirically described by both power-law and exponential growth curves (Fig. 7). Power-law acceleration of the strain rate, earthquakes and other precursors have been widely reported before material-failure phenomena, including volcanic eruptions[58–60]. The characteristic failure time, $t_f$, for the power-law relationship is ∼3,900 days for Rabaul and ∼1,240 days for Sierra Negra (where the volcanoes erupted 3,100 and 950 days after the beginning of the anomaly, respectively). For CFc, similar calculations yield a failure time of 5,670 ± 735 days from the beginning of the anomaly (that is, 2005). Characteristic times, $t_*$, of the exponential curve of 240, 850 and 1,600 days are obtained for Sierra Negra, Rabaul and CFc, respectively. The failure time cannot be defined in the exponential model[59], but the Rabaul and Sierra Negra examples show that eruptions occurred at 3–4 $t_*$ (Fig. 7).

However, the temporal evolution of volcanic unrests often exhibits complex, nonlinear behaviour, such as in the case of the Yellowstone caldera, where an initial exponential/power-law acceleration of ground uplift during 2004–2008 was followed by a deceleration[61] (Fig. 7). A pausing of ground uplift occurred at CFc between 1982–1984 and 2005. Caution is therefore prudent when forecasting the future mid- to short-term evolution of any period of unrest. Even if the magma underneath CFc is likely to be approaching the CDP, the possible future scenarios can be complicated by additional processes that have not been considered here. For example, increases in the melt liquidus due to H$_2$O release and consequent magma crystallization could increase the melt viscosity, and therefore act against further magma migration. Additional careful scrutiny of monitoring data in the coming months and years is key to interpreting whether hydrothermal heating or magma quenching will prevail.

## Methods
**Data set.** Data on the compositions of fumarole emissions as well as the analytical methods used in this study and their uncertainties are available in the literature[14]. The used data set was updated to December 2015 (Supplementary Data 1).

Fifteen continuous-monitoring GPS stations (CGPS) operate at CFc. A full description of the GPS network and the complete data set of the displacement time series is available elsewhere[62]. Here we used the vertical displacement time series of the RITE CGPS station, which is located in the Pozzuoli area (Fig. 3a) where the uplift has been greatest over the past 15 years. Data on the numbers and magnitudes of earthquakes were obtained from the literature[14,20] and from INGV Osservatorio Vesuviano. The vertical displacements at Rabaul, Sierra Negra, and Yellowstone were also taken from the literature[16,57,61].

**Physical modelling of magmatic gas injection into the hydrothermal system.** Our physical model aims at assessing the thermal effects caused by the IMF into the CFc hydrothermal system. Among the numerous possible scenarios, we select two reference cases.

In the first case (Simulation 1) we considered an input magmatic fluid of fixed (that is, time invariant) composition, with a CO$_2$/H$_2$O molar ratio of 0.67. This composition corresponds to that of a gas in equilibrium at 200 MPa with a trachybasalt magma (H$_2$O = 3.39 wt%, CO$_2$ = 0.079 wt%; blue circle in Fig. 1). This CO$_2$-rich composition (CO$_2$ molar fraction = 0.4) was inferred based on volcanic gas evidence[50] and has been used in previous CFc simulations[13].

In the second scenario (Simulation 2) we injected the virtual hydrothermal system with a magmatic gas that has a composition that becomes richer in H$_2$O over time (Supplementary Table 1). This evolving gas composition is based on the output of open-system magma-degassing models of CFc magmas. A comparison of measured fumarole emissions and the model degassing trends[21] provides support for open-system magma degassing being an ongoing process at CFc. The results of model simulations[21] (Supplementary Fig. 4a) demonstrate that gas observations (in the N$_2$–He–CO$_2$ gas system) can only be quantitatively reproduced if open-system decompression is assumed to be present. We consider the case of a trachybasalt magma decompressing from 200 to 130 MPa, initially coexisting with a gas phase having N$_2$/He = 900 and N$_2$/CO$_2$ = 0.0047 (Supplementary Fig. 4a). We reconstruct the temporal evolution of the CO$_2$/H$_2$O ratio of the magmatic gas feeding the CFc hydrothermal system (Supplementary Fig. 5b) from that of the N$_2$/He ratio measured in CFc fumaroles (Supplementary Fig. 5a). Fumarole N$_2$/He ratios are converted into CO$_2$/H$_2$O ratios in the source magmatic gas using the

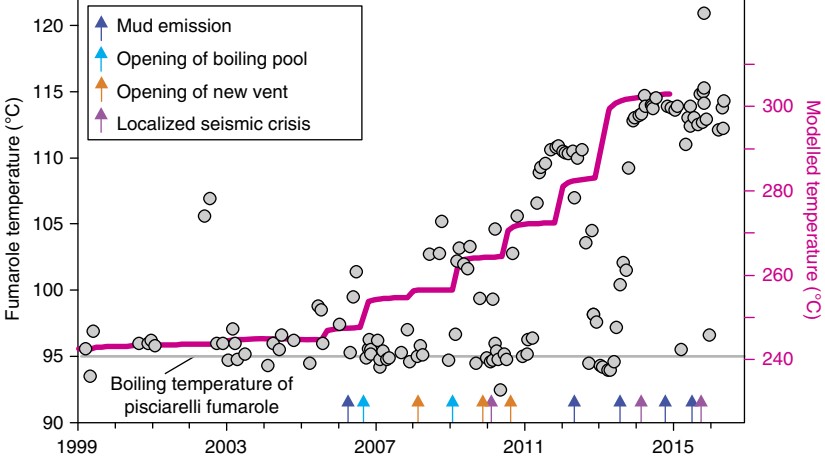

**Figure 6 | Discharge temperature at the Pisciarelli fumarole compared with the modelled temperature.** The temperature at the Pisciarelli fumarole (gray dots) increased from the boiling temperature (95 °C) in 2005–2006 to 115–120 °C in 2015. During the same time interval, temperature increased of only 3–4 °C at the highest temperature fumarole BG, implying clustering of hydrothermal influx on the eastern outer slope of Solfatara crater, where Pisciarelli is sited (Fig. 3). At Pisciarelli, localized low-magnitude seismic swarms[20], a weak phreatic activity (mud emission, opening of boiling pools and new vents), and a strong increase in the fumarole flow rate accompanied the temperature increase. The variation occurs concurrently with the increasing temperature of the CFc hydrothermal system modelled in Simulation 2 (magenta line; that is, average temperature inside the yellow box in Fig. 4).

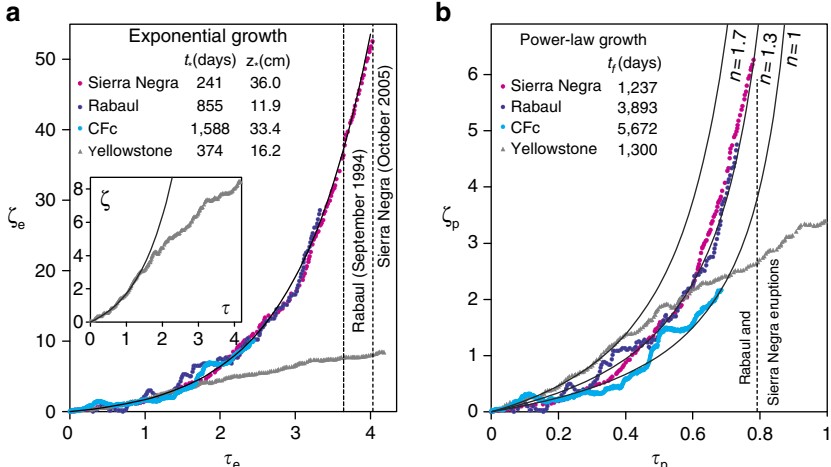

**Figure 7 | Examples of several-years-long accelerating deformation patterns at selected key volcanoes.** The curves indicate the vertical ground displacement observed at the different volcanoes (Sierra Negra magenta, Rabaul blue, CFc cyan, Yellowstone gray; see Methods). The patterns can be fitted with either exponential (**a**) or power-law (**b**) growth curves. (**a**) The dimensionless time ($\tau_e$) and vertical displacement ($\zeta_e$) are given by $\tau_e = t/t_*$ and $\zeta_e = z/z_*$, respectively, where $t_*$ and $z_*$ are the characteristic time and vertical displacement, respectively, of the exponential curve: $z(t) = z_* \, [exp(t/t_*) - 1]$. (**b**) The dimensionless time ($\tau_p$) and vertical displacement ($\zeta_p$) are given by $\tau_p = t/t_f$ and $\zeta_p = z/\left(C/t_f^n\right)$, respectively, where $t_f$ C, and $n$ the adjustable parameters of the power-law curve: $z(t) = C/(t_f - t)^n - C/t_f^n$. The initial time for each case refers to the first evidence of uplift. Different curve parameters are reported in the figure legends; the values of $C$ ($10^4 \times cm \times days^n$) are 43.0, 33.8, 12.0, 67.2 for Sierra Negra, Rabaul, CFc, and Yellowstone, respectively.

model relationship in Supplementary Fig. 4a. The obtained time dependence of the $CO_2/H_2O$ ratio (see red line in Supplementary Fig. 5b) was sampled at specific time intervals to infer the magmatic gas $CO_2/H_2O$ ratio composition of each IMF event (Supplementary Table 1). The degassing decompression trend (that is, from 200 to 130 MPa) of this model is schematically illustrated by the orange lines in Fig. 1a,d.

The energy transported by volatiles while they are degassed from magma (Fig. 1b) was calculated from the enthalpy of the separated $H_2O$–$CO_2$ mixture at the magma temperature and pressure. The method[63,64] we used can simulate properties of a binary $CO_2$–$H_2O$ mixture up to supercritical conditions. However, we injected magmatic fluids into the virtual hydrothermal system at 350 °C in both Simulations 1 and 2. This is because the geothermal simulator that we used (TOUGH2 (ref. 26)) can only work below the critical-point temperature of water. Enthalpies of the magmatic gas phase at 350 °C were calculated using the EOS2 module[26]. This yielded that the enthalpies of the injected magmatic gases were far lower than those of magma-released supercritical fluids (Fig. 1b). This difference accounts for the energy lost by fluids as they pass thorough the plastic zone, between the magma and the overlying hydrothermal system.

TOUGH2 is a numerical simulator for geothermal systems that is fully described elsewhere[26]. Briefly, TOUGH2 includes the coupled transport of a multiphase (gas/liquid) fluid and heat through a porous medium. The TOUGH2 version used here takes into account the contemporaneous presence of two fluid components ($H_2O$ and $CO_2$) at subcritical temperature conditions[26]. More-recent geothermal simulators[65] can also work at supercritical temperature conditions, but we used TOUGH2 to allow comparisons with previous work at CFc[13,66,67]. The model includes heat flow occurring by both conduction and convection, with the latter including both sensible and latent heat effects[26]. The model does not account for either the presence of other gas components, solute transport, and chemical reactions, or for deformation or fracturing of the porous rock.

The physical model calculations included defining the computational domain, achieving steady-state conditions before simulating IMF events, selecting the number and timing of IMF events, and calculating the mass of fluids injected in each IMF event. These steps are described in detail below.

The computational domain consists of 850 cells distributed in a two-dimensional radial space of 2,000 m × 2,500 m (Fig. 4). This domain is similar to (but 500 m deeper than) that used in previous simulations[66–69] of the CFc hydrothermal system[13]. The top boundary is at fixed (atmospheric) temperature and pressure conditions, while other boundaries are impermeable and adiabatic.

As in previous work[13], the rock physical properties are homogeneous. The fluid source is located at the base of the domain (Fig. 4).

The second step consisted in achieving steady-state conditions. Initially the domain contains liquid water with a low $P_{CO_2}$ ($CO_2$ partial pressure) and a temperature that varies with depth according to the thermal gradient measured in CFc geothermal wells[67]. This condition is perturbed with a 2000-year-long injection of a 350 °C gas mixture at 39 kg s$^{-1}$, with a $CO_2/H_2O$ molar ratio of 0.17. The flux and composition reflect measurements performed at Solfatara[50,67]. The system reaches a steady-state condition after 2000 years of injection, which represents time 0 of Simulations 1 and 2. Ideally this initial condition represents the state of the system before 30 October 1983, which is the date on which magmatic gas was first injected into the base of the system (Supplementary Fig. 1). Under steady-state conditions, the model predicts the formation of a gas zone (Xg = 1) at 200–220 °C at the top of the domain, which is aligned with the injection zone and just below the Solfatara fumarole field ('checkpoint for gas composition' in Fig. 4).

In the third step, we simulated 14 IMF events. Their timing (Supplementary Table 1) was inferred based on compositional anomalies that have been determined for Solfatara fumaroles. The first three events occurred during the major bradyseismic crisis in 1983–1984, and two minor uplift episodes occurred in 1989 and 1994 (ref. 66). Peaks in the $CO_2/H_2O$ ratio at Solfatara fumaroles (Supplementary Fig. 1b) marked each event. Successive peaks (from 2000 to 2015) in $CH_4$-based geoindicators (that is, the fumarole ratios of $CO_2/CH_4$ and $He/CH_4$ (refs 14,70)) are taken as evidence for the occurrence of 11 IMF episodes (Supplementary Figs 1a and 2b). These events typically caused both pulsed inflations of the terrain[13,14] (Supplementary Fig. 2c) and localized seismic activity (Supplementary Figs 1f and 2d,e).

Finally, the temporal evolution of the fumarole emissions of $H_2O–CO_2$ (Supplementary Fig. 1b) can be used to confine the total amount of fluid injected in each of the 14 IMF events (Supplementary Table 1) using a trial-and-error method. The total mass of each IMF is adjusted until the modelled compositions in the 'checkpoint for gas composition' fit the observed fumarole $CO_2/H_2O$ ratio reasonably well (Supplementary Fig. 1b). As explained above, the $CO_2/H_2O$ molar ratio of the injected fluids was assumed to be constant ( = 0.67) in Simulation 1, while it decreased from 0.67 to 0.22 in Simulation 2, as in an open-system magma-degassing process (Supplementary Fig. 5b; Supplementary Table 1).

The simulation outputs relevant to this study are the simulated values of the total pressure, temperature, Xg (volumetric gas fraction), $P_{CO_2}$, and $X_{CO_2}$ ($CO_2$ mass fraction) in each of the 850 cells. A key model output is the temporal evolution of the system temperature during the injection of the magmatic fluids (Fig. 5a). This corresponds to the volume-weighted average temperature of the cells in a cylinder with a diameter of 0.95 km and a height of 1 km (that is, volume of 0.707 km$^3$) sited above the injection zone ('temperature box' in Fig. 4).

**Data availability.** All relevant data are available from the authors.

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

## Acknowledgements

This work received funding from the INGV-DPC Research Agreement 2012–2015, the EU-FP7 Project MED-SUV (grant agreement 308665), the ERC Project 'Bridge' (grant agreement 305377), the INGV project COHESO, and DECADE (an initiative from the Deep Carbon Observatory). Comments from V.C. Smith improved the quality of the manuscript. We thank the Seismology Laboratory of INGV Osservatorio Vesuviano for providing the list of CFc earthquakes. We wish to thank Y. Taran, L. Karlstrom and an anonymous reviewer for the helpful suggestions which improved the quality and the clarity of the manuscript.

## Author contributions

G.C. conceived the initial idea of the study, with all of the coauthors defining the methodology and strategy. G.C., S.C. and P.D.M. acquired geochemical and geodetic data. G.C. ran the simulations, with contributions from A.P. and A.C. G.C., A.P., A.A., A.C., J.V. and V.A. wrote the manuscript with input from all of the coauthors.

## Additional information

**Competing financial interests:** The authors declare no competing financial interests.

