## [Peer Review File · Nature Communications]

Reviewers' comments:

Reviewer #1 (Remarks to the Author):

This article is a multidisciplinary approach to a complex problem and as such provides a unique set of observations that lead to compelling conclusions that are, to say the least, exciting in their scope. The authors show how simple models for volatile saturation can be used to extrapolate thermal perturbations in a hydrothermal system. They then conclude that this is a dominant control on ground deformation in these large hydrothermal systems and go some way to showing that these systems often accelerate in terms of deformation observables toward critical failure and eruption points. In this way, the work is provocative as they do not shy from predicting those critical points for the CFc. It is my opinion that this article reaches the criteria necessary for publication in a high impact journal such as Nature Communications, however, in the interests of the best quality, I have some suggested revisions (below).

First, however, I confirm that the findings are novel and interesting, the conclusions are valid and supported by the manuscript and with some minor exceptions, the work is clear.

(1) As tends to be the case with multidisciplinary models that provide a synthesis of data in an overarching text, this manuscript lacks some detail. Even though it might be well established in other pieces of work, it is essential for clarity that the key steps are laid out for the reader. For example, how do the authors go from degassing models to energies and temperatures? The extension of this work from a degassing study to a temperature-dependent uplift study hinges on this conversion so it is worth stating it plainly by including an in-line equation. It is essential that the critical computations are given in a transparent way (in the main text or the methods). As a reviewer I am unable to reproduce anything in this piece of work. For the most sceptical scientist, this smacks of hiding things and leaving work opaque and unclear.

(2) While Figure 4 is interesting, it does not follow from the previous figures. Unless I am mistaken, the ground deformation data given in Figure 4 are from GPS data at CFc (Used data section in the Methods). If this is so then these results do not come from the degassing model itself. Therefore, the results presented in Figure 4 is a separate analysis from what comes before. In this case, I would highly encourage the authors to structure the introductory paragraphs to explain that what will follow is a two-step approach in which they will establish (1) that the critical degassing pressure is reached in multi-fluid saturation models which controls thermal input in hydrothermal systems; and (2) that the ground deformation accelerates toward critical points. Then, finally, the conclusion is simply that these two processes may be related at CFc and elsewhere. I do understand that the relationship between (1) and (2) is proposed through the thermal expansion calculation shown in Figure 3, however, in my opinion much stronger conceptual links must be made between the first and second finding of this work.

(3) In Figure 4, the authors show that both exponential and power law trends can fit these data. An apt citation here might be the recent work of Vasseur et al., (2015) who go to lengths to show that this difference is associated with the porosity of the magma or rock.

Also in reference to Figure 4. I find this a good example of the opacity of this piece of work. The Figure caption includes some unreferenced, equations (thankfully something to go on) but not all the parameters are explained. Have these forms been used elsewhere? These are not the exponential or power law forms used in the papers cited (Bell et al., 2011; Voigt, 1988). If I had to guess (and I do have to guess) these are simply examples of a normalized power law and normalized exponential growth laws but that they do not contain any idea of the mechanisms involved in the growth (whereas

Voigt, 1988 makes this effort to link mechanics with the growth). If this is the case, it should be clearly and honestly stated that these are purely empirical examples of growth laws of a general form for which vertical displacement and time are non-dimensionalized. I consider it dishonest to present this analysis as if it is robust (especially when great work is available that tests these growth laws against many eruption precursors).

(4) Throughout the manuscript the written English is sometimes awkward and needs a careful review by the authors before resubmitting.

(5) Often acronyms are used but are not always repeated. This leads to a false impression of complexity and smears clarity. Example: CI for Campanian Ignimbrite.

(6) Axis labels in all figures are not clear. Clear figure making is essential. Figure 2b, as an example, is hard to read when "depth" is not given as the y-axis.

(7) Please remove reference (after Figure 4) to the conclusion that this manuscript has "clearly demonstrated" any potential hazards for anything. This manuscript was not dedicated to hazard demonstration. Rather, it has demonstrated some mechanistic origin of ground deformation preceding large eruptions. There's no need to overreach. The work is excellent for what it is.

(8) In my opinion, the Methods section needs expanding. This would be a stunning piece of work if short summaries of each technique were given. For example, what are the limits of applicability of Tough2? How is that simulation structured? What does it compute specifically and on what grounds? What are the governing equations? These are questions that really interested me and I was disappointed not to find the answer in the methods. Of course, I am directed to the appropriate article where I can find this information, but it doesn't hurt to have it here also, albeit in summary.

(9) Finally, the title is not a good choice as it is not cognate with the content.

Reviewer #2 (Remarks to the Author):

Review of manuscript NCOMMS-16-06483-T by G.Chiodini et al.

I read this manuscript with a great interest. The paper considers a new idea about an abrupt increase in the amount of the degassed H₂O from a silicate melt at decreasing pressure, when almost all CO₂ is already degassed. This idea is supported by modeling based on known solubilities of the H₂O+CO₂ mixture in a trachy-andesitic melt. The sharp increase of the H₂O fraction in the releasing from magma fluid drastically increases the thermal capacity of the fluid and consequently, heats the overlying hydrothermal reservoir. According to the authors, in that case a critical situation can appear that can eventually cause the eruption. This approach is applied to the world best studied case of the Campi Flegrei volcanic centre using an unprecedented data set of the chemical composition of fumaroles in Solfatara crater. Almost ideal correlation between N₂/He in the Solfatara geothermal gases during more than 30 years of observations and the uplift of the caldera floor that has been demonstrated in a recent study by Chiodini et al. (2015), has allowed to constrain variations in the pressure of the magma degassing and to build a strong conceptual model for the Campi Flegrei unrest, where chemical data strongly support the observed geophysical events. The TOUGH modeling showed how the temperature of the main hydrothermal reservoir can rise depending on the initial H₂O/CO₂ in the degassing melt and at changing H₂O/CO₂ ratio. Finally, the apparent power-law fitting the deformation data made possible a crude prognosis of a possible eruption based on the material failure phenomena, that has been successfully applied for other cases by other authors.

In my opinion, this paper is almost ready to be accepted. Just few comments along the text.

I guess, authors have to clearly define what they mean when say about "open system degassing". An open system can be with supplying the releasing gases from depth, or to be an infinite reservoir, or as Rayleigh process from a constant volume. It is also important that this open system degassing reservoir should be (?) well (ideally) mixed.

P.2 Define "critical state"

"heat transfer to overlying rocks and hydrothermal system will exhibit complex (non-linear) behavior"
- Heat transfer is non-linear by definition. Thermal conductivity equation is non-linear .

No need to refer Giggenbach (1996) speaking about inert or poorly reactive gases.

P.5 "...large variations in N₂-CO₂-He-Ar..". You do not discuss Ar, N₂/Ar, Ar isotopes, as well as H₂O/CO₂ and water isotopes. And you did not touch the issue about water origin, as well as the correspondence between modeled increase in H₂O/CO₂ in releasing magmatic fluid and the observed decrease in H₂O/CO₂ in hydrothermal gases.

Fig. 3 Explain what sense in the scale change between modeled and CO/CO₂ temperatures

The uplift for Yellowstone is rather linear in contrast to exponential or power-law for other volcanoes.

Yuri Taran

Reviewer #3 (Remarks to the Author):

This manuscript proposes the idea that distinct solubilities of H₂O and CO₂, the two primary volatile species in magmas, lead to distinct open-system volatile release as a function of pressure, and that the transition to dominant H₂O exsolution during gradual decompression may control volcanic unrest and eruption. This work builds on previous studies by these authors on deformation and hydrothermal phenomena associated with recent unrest at Campi Flegrei, although there is discussion of other volcanoes. There are some interesting ideas in here that are worthy of publication in Nature. My main comments as detailed below relate to whether the connection of ideas to data is compelling. I will also note that there are slight English language issues that need to be addressed.

The concept of a critical degassing pressure makes sense. It would seem that more justification of the open system assumption is warranted. You need to get bubbles out of the magma chamber, and this introduces a timescale that must be met (bubble rise, and creep closure in mushy wall rocks) that may depend on magma composition, thermal state and depth.

Doesn't the release of magmatic gases also result in precipitation of hydrous minerals around the magmatic system? Does this affect the mass you predict? What amount of heat would this involve.

Regarding the application to Campi Flegrei. It would seem that some assessment of model uncertainty is required. Do the results change if permeability values or ratio of horizontal/vertical change? I understand that these are tuned to match the observed lag between ground deformation and fumerole emission, but the delay should depend on what the source of deformation is (expansion due to heating in the subsurface or magmatic source) and what the rheology is (viscoelastic/plastic effects such as

are discussed here would modify the lag). I expect that such considerations are important for the total amount of gas released as well.

Of course the relationship between pressure change and volume change depends on the fraction of bubbles in the chamber (e.g., Huppert and Woods 2002 JGR, also see recent work by Edmonds et al on different roles of H₂O vs CO₂). I see no discussion of this in relation to parameter estimation, but such would seem to be critical. Given the different heating capacities, it also would be good to place constraints on results due to CO₂/H₂O ratio uncertainties.

What constrains the background temperature field? Is steady state a good assumption? It would also seem that the results will be highly sensitive to the background temperature field in the crust.

Finally, the exponential and power law fits to Rabaul, Sierra Negra, Campi Legrei and Yellowstone are very thought provoking. But it would be good to provide more details about how the deformation signals were compiled: is this vertical displacement averaged over some area? Presumably there is some uncertainty in the timing of first uplift? How does volatile composition/quantity affect this trend?

Point-to-point response to reviewers' comments (authors' replies in red)

Reviewer #1

This article is a multidisciplinary approach to a complex problem and as such provides a unique set of observations that lead to compelling conclusions that are, to say the least, exciting in their scope. The authors show how simple models for volatile saturation can be used to extrapolate thermal perturbations in a hydrothermal system. They then conclude that this is a dominant control on ground deformation in these large hydrothermal systems and go some way to showing that these systems often accelerate in terms of deformation observables toward critical failure and eruption points. In this way, the work is provocative as they do not shy from predicting those critical points for the CFC. It is my opinion that this article reaches the criteria necessary for publication in a high impact journal such as Nature Communications, however, in the interests of the best quality, I have some suggested revisions (below). First, however, I confirm that the findings are novel and interesting, the conclusions are valid and supported by the manuscript and with some minor exceptions, the work is clear.

We thank the reviewer for the positive comments.

(1) As tends to be the case with multidisciplinary models that provide a synthesis of data in an overarching text, this manuscript lacks some detail. Even though it might be well established in other pieces of work, it is essential for clarity that the key steps are laid out for the reader. For example, how do the authors go from degassing models to energies and temperatures? The extension of this work from a degassing study to a temperature-dependent uplift study hinges on this conversion so it is worth stating it plainly by including an in-line equation. It is essential that the critical computations are given in a transparent way (in the main text or the methods). As a reviewer I am unable to reproduce anything in this piece of work. For the most sceptical scientist, this smacks of hiding things and leaving work opaque and unclear.

We have fully accepted the reviewer's invitation to clarify the methodology that we used to obtain the results described in the manuscript. With this aim, we have revised and expanded the section entitled "*Physical modeling of magmatic gas injection into the hydrothermal system, and the steps in the model calculations*" in the Methods so as to clarify the most critical points highlighted by the reviewer. Other parts of the Methods section have been restructured to provide a more detailed description of the computational sequence.

In detail, to address the specific point made by reviewer#1 we have clarified in the Methods the procedure used to calculate the energy associated with the magmatic H₂O-CO₂ mixtures (Fig. 1b). The new text reads as follows: "...The energy transported by volatiles while they are degassed from magma (Fig. 1b) was calculated from the enthalpy of the separated H₂O-CO₂ mixture at the magma temperature and pressure. The method {Afanasyev, 2012 #47; Afanasyev, 2013 #8} we used can simulate properties of a binary CO₂-H₂O mixture up to supercritical conditions..." In brief, we used a published equation of state for H₂O-CO₂ mixtures that can apply over large temperature and pressure ranges. The multiphase thermodynamic equilibrium in a H₂O-CO₂ binary mixture is expressed in pressure–enthalpy–mixture-composition variables, in contrast to classical thermodynamic pressure–temperature–composition variables. It has been reported {Afanasyev, 2012 #47; Afanasyev, 2013 #8} that the method can be used to describe real properties of mixtures over wide ranges of pressures and temperatures containing critical points. The computation is complex and not explainable in few sentences, and so we point the readers to the article in which it is described in detail (Ref. 38; Afanasyev, A. A. Simulation of the properties of a binary carbon dioxide–water mixture under sub- and supercritical conditions. High Temperature 2012; 50 (3), 340–347). The computations were performed with an equation of state included in MUFITS, which is noncommercial software that is described in another citation (Ref. 39; Afanasyev, A. A. Application of the reservoir simulator MUFITS for 3D modelling of CO₂ storage in geological formations. Energy Procedia 2013; 40, 365–374).

The second specific point of the reviewer asking about "how [we] go from degassing models to energies and temperatures..." refers to our simulations of the interactions of magmatic fluids with the hydrothermal system into which they are injected. The simulations were performed using TOUGH2, which is a very well

known geothermal simulator in the geothermal and volcanological communities (several hundred published papers refer to this simulator). We accept the comment made by the reviewer that this part in the original paper was probably not sufficiently clear. We have therefore almost completely rewritten one of the parts of the Methods section, entitled “*Physical modeling of magmatic gas injection into the hydrothermal system, and the steps in the model calculations*”. We now detail the strategy and each step in the model calculations. Furthermore, we have provided the input information for the computations. In particular, in the Supplementary Material S2 we give the initial values of pressure, temperature, and PCO_2 (carbon dioxide partial pressure) in each cell of the domain. Note that this information, along with the sequence of magmatic fluid injections (Extended Data Table 1) and the descriptions reported in the Methods, is sufficient to allow anyone to repeat our simulations using TOUGH2.

(2) While Figure 4 is interesting, it does not follow from the previous figures. Unless I am mistaken, the ground deformation data given in Figure 4 are from GPS data at CFC (Used data section in the Methods). If this is so then these results do not come from the degassing model itself. Therefore, the results presented in Figure 4 is a separate analysis from what comes before. In this case, I would highly encourage the authors to structure the introductory paragraphs to explain that what will follow is a two-step approach in which they will establish (1) that the critical degassing pressure is reached in multi-fluid saturation models which controls thermal input in hydrothermal systems; and (2) that the ground deformation accelerates toward critical points. Then, finally, the conclusion is simply that these two processes may be related at CFC and elsewhere. I do understand that the relationship between (1) and (2) is proposed through the thermal expansion calculation shown in Figure 3, however, in my opinion much stronger conceptual links must be made between the first and second finding of this work.

We thank the reviewer for this comment that helped us to improve the clarity of the manuscript. We have reworked the introductory paragraphs following the argument flow suggested by the reviewer as follows: “*The present study linked (i) magma degassing at depth with (ii) the resulting perturbation in the overlying hydrothermal system. Here we initially use the results of volatile saturation {Papale, 2006 #26} models to demonstrate that decompressing magmas can reach a critical condition, which we refer as a critical degassing pressure (CDP), at which their ability to release water and convectively transport heat is increased by a least an order of magnitude. We then use physical models {Pruess, 1991 #27} to show that magmatic volatiles released at the CDP, when injected into an overlying hydrothermal system, lead to extensive heating and expansion, and cause temporally accelerating ground deformation. Finally, we examine ground deformation time series from CFC and some other restless calderas, which identifies consistent accelerating ground uplift trends that are reminiscent of those predicted by our model. We conclude that magmas at the CDP can be recurrent causal factors in driving volcanic unrest toward a critical state; that is, a state near a bifurcation at which the system can evolve either culminating in an eruption or cooling down (REF, e.g. Tilling).*” Moreover, we have added a few sentences to the Discussion section to better explain the link between the heating of the system and the ground deformation (see references to Heap and coworkers and to Desseault and coworkers).

(3) In Figure 4, the authors show that both exponential and power law trends can fit these data. An apt citation here might be the recent work of Vasseur et al., (2015) who go to lengths to show that this difference is associated with the porosity of the magma or rock. Also in reference to Figure 4. I find this a good example of the opacity of this piece of work. The Figure caption includes some unreferenced, equations (thankfully something to go on) but not all the parameters are explained. Have these forms been used elsewhere? These are not the exponential or power law forms used in the papers cited (Bell et al., 2011; Voigt, 1988). If I had to guess (and I do have to guess) these are simply examples of a normalized power law and normalized exponential growth laws but that they do not contain any idea of the mechanisms involved in the growth (whereas Voigt, 1988 makes this effort to link mechanics with the growth). If this is the case, it should be clearly and honestly stated that these are purely empirical examples of growth laws of a general form for

which vertical displacement and time are non-dimensionalized. I consider it dishonest to present this analysis as if it is robust (especially when great work is available that tests these growth laws against many eruption precursors).

We think that some of the comments made by the reviewer are due to misinterpretation of what we tried to explain in a very concise way (due to space limitations). We never meant to apply the Voigt (1988) model (or its more recent versions) to infer the eruption time, or to explain the underlying mechanics. For the sake of brevity, we cited the two references to introduce the power-law model. However, the relationship we used is a solution of the Voigt equation applied to the time evolution of vertical ground deformation for the studied systems, and the relationships reported in the figure are made dimensionless by scaling by the corresponding characteristic parameters; that is, the scaling is not arbitrary. In the revised version, as suggested by the reviewer, we refer to the recent study of Vasseur et al. (2015), who introduced both the power-law and exponential models. In addition, we have improved the descriptions of all of the parameters, and clearly state that our ground deformation analysis is purely empirical. Moreover, the used equations are reported explicitly in the figure caption, together with all of the characteristic parameters.

(4) Throughout the manuscript the written English is sometimes awkward and needs a careful review by the authors before resubmitting.

The manuscript has been edited thoroughly by a native English speaker working for a professional English editing service.

(5) Often acronyms are used but are not always repeated. This leads to a false impression of complexity and smears clarity. Example: CI for Campanian Ignimbrite.

All of the unnecessary acronyms have been removed.

(6) Axis labels in all figures are not clear. Clear figure making is essential. Figure 2b, as an example, is hard to read when "depth" is not given as the y-axis.

We have corrected Fig. 2b and the other figures where necessary.

(7) Please remove reference (after Figure 4) to the conclusion that this manuscript has "clearly demonstrated" any potential hazards for anything. This manuscript was not dedicated to hazard demonstration. Rather, it has demonstrated some mechanistic origin of ground deformation preceding large eruptions. There's no need to overreach. The work is excellent for what it is.

Thank you for the comment; we have removed that sentence.

(8) In my opinion, the Methods section needs expanding. This would be a stunning piece of work if short summaries of each technique were given. For example, what are the limits of applicability of Tough2? How is that simulation structured? What does it compute specifically and on what grounds? What are the governing equations? These are questions that really interested me and I was disappointed not to find the answer in the methods. Of course, I am directed to the appropriate article where I can find this information, but it doesn't hurt to have it here also, albeit in summary.

We have expanded and substantially revised the Methods section in order to address the criticisms made by the reviewer (see above).

(9) Finally, the title is not a good choice as it is not cognate with the content.

We have changed the title as follows: "Magmas near the critical degassing pressure drive volcanic unrest toward the critical state."

Reviewer #2

I read this manuscript with a great interest. The paper considers a new idea about an abrupt increase in the amount of the degassed H₂O from a silicate melt at decreasing pressure, when almost all CO₂ is already degassed. This idea is supported by modeling based on known solubilities of the H₂O+CO₂ mixture in a trachy-andesitic melt. The sharp increase of the H₂O fraction in the releasing from magma fluid drastically increases the thermal capacity of the fluid and consequently, heats the overlying hydrothermal reservoir. According to the authors, in that case a critical situation can appear that can eventually cause the eruption. This approach is applied to the world best studied case of the Campi Flegrei volcanic centre using an unprecedented data set of the chemical composition of fumaroles in Solfatara crater. Almost ideal correlation between N₂/He in the Solfatara geothermal gases during more than 30 years of observations and the uplift of the caldera floor that has been demonstrated in a recent study by Chiodini et al. (2015), has allowed to constrain variations in the pressure of the magma degassing and to build a strong conceptual model for the Campi Flegrei unrest, where chemical data strongly support the observed geophysical events. The TOUGH modeling showed how the temperature of the main hydrothermal reservoir can rise depending on the initial H₂O/CO₂ in the degassing melt and at changing H₂O/CO₂ ratio. Finally, the apparent power-law fitting the deformation data made possible a crude prognosis of a possible eruption based on the material failure phenomena, that has been successfully applied for other cases by other authors.

In my opinion, this paper is almost ready to be accepted. Just few comments along the text.

We thank the reviewer for the positive comments.

I guess, authors have to clearly define what they mean when say about "open system degassing". An open system can be with supplying the releasing gases from depth, or to be an infinite reservoir, or as Rayleigh process from a constant volume. It is also important that this open system degassing reservoir should be (?) well (ideally) mixed.

Open-system degassing is modeled as a classical Rayleigh process, as explicitly stated in the revised version of the manuscript, in the caption of Fig. 1: "*...an open-system Rayleigh-type degassing process (where at each infinitesimal decompression step, an infinitesimal parcel of gas phase in excess of the permissible saturation is distilled from the well-mixed magma...)*." However, we stress that results virtually identical to those shown in Fig. 1 would be obtained with an open-system multistep degassing process for the following reason (as stated in the revised text): "*...multistep degassing can be adequately reproduced as an open-system degassing process provided that there are numerous and recurrent system-opening events, as is likely to be the case.*" Open-system magma degassing is strongly supported by the composition of the fumarole emissions. Caliro et al. (2015) demonstrated that the long-term evolution of the N₂-He-CO₂-Ar compositions can only be reproduced in open-system conditions (we have added a figure comparing the observed and theoretical compositions; see Extended Data Fig. 3), while a closed system is not able to reproduce the observations at all. Moreover, an open system is required to account for the large amount of volcanic fluids emitted by both the fumaroles and diffuse degassing at Solfatara. Open-system degassing processes—either continuous or multistep—seem to be a common feature beneath volcanoes, and have often been used to quantitatively explain compositional (inert gas) changes at several volcanic systems (e.g., Vulcano, Etna, worldwide MORB systems, and Planchón-Peteroa). We argue in the manuscript that the prevalence of open-system volcano behaviors may be strictly related to the geochemistry of magmatic plumbing systems, composed of an intricate network of dikes, sills, and small chambers with irregular shapes that extend upward through the crust (Preston, 2001; Dawson et al., 2004; Cartwright and Hansen, 2006; Paulatto et al., 2010). Such geometries can act as traps where magma ascent is inhibited and gas–melt decoupling can occur, guaranteeing open-system degassing.

In the revised version of the manuscript we have included a discussion on closed- vs. open-system degassing: "*...our calculations were performed under open-system conditions since the long-lasting variations in the fumarole-gas composition at CFC cannot be reproduced in a closed system, instead requiring efficient*

separation of gases from the magma²¹. The large amount of magmatic fluids released by CFC manifestations⁴¹ also supports an ongoing open (rather than closed) magma degassing behavior...” We further argue that open-system degassing “... can result from the complex geometry of crustal volcano plumbing systems, whose intricate networks of fractures, dikes, sills, and small reservoirs⁴³⁻⁴⁶ facilitate the segregation of gas from melt, and the loss of volatiles from a foam layer⁴⁷. Foam growth in low-viscosity mafic melts takes place over timescales of months to a few years^{47, 48}, which is faster than the observed decennial trends in gas composition...”

P.2 Define "critical state"

We have defined “critical state” in the last sentence of the Introduction: “...*We conclude that magmas at the CDP can be recurrent causal factors in driving volcanic unrest toward the critical state; that is, in a state near a bifurcation at which the system can evolve either culminating in an eruption or cooling down (REF, e.g. Tilling).*”

"heat transfer to overlying rocks and hydrothermal system will exhibit complex (non-linear) behavior" - Heat transfer is non-linear by definition. Thermal conductivity equation is non-linear .

In order to avoid confusion we have deleted the term “nonlinear”: “...*the pattern of heat transfer to overlying rocks and hydrothermal systems will be complex and will vary as the unrest progresses...*” The meaning of this sentence is clear in Fig. 1b.

No need to refer Giggenbach (1996) speaking about inert or poorly reactive gases.

We have deleted the reference.

P.5 “..large variations in N₂-CO₂-He-Ar..”. You do not discuss Ar, N₂/Ar, Ar isotopes, as well as H₂O/CO₂ and water isotopes. And you did not touch the issue about water origin, as well as the correspondence between modeled increase in H₂O/CO₂ in releasing magmatic fluid and the observed decrease in H₂O/CO₂ in hydrothermal gases.

Here we are referring to the following recently published study that is cited at the end of the sentence: “(Caliro, S., Chiodini, G. & Paonita, A. *Geochemical evidence of magma dynamics at Campi Flegrei (Italy)*. *Geochim. Cosmochim. Acta* 2014; 132, 1–15).” That study addressed in detail the large variations observed in the N₂-CO₂-He-Ar gas system. In order to clarify the use of this reference, we moved the citation to another part of the text: “...*The large variations in the fumarole emissions of N₂-He-CO₂-Ar²¹...*”

We have added a sentence about the origin of the water origin to the caption of Fig. 3, based on the results for the stable isotopes: “... *Previous geochemical investigations based on the stable isotopes of water revealed the presence of typical magmatic waters in the Solfatara fumarole vents ...*”

We discuss the “modeled increase in H₂O/CO₂” vs. “the observed decrease in H₂O/CO₂ in hydrothermal gases” in the last sentence of the Results section: “...*One interesting outcome of Simulation 2 is that, while the CO₂/H₂O ratio of the injected magmatic fluids decreases over time, the simulated gas composition at the “checkpoint for gas composition” (Fig. 4) becomes increasingly rich in CO₂ (Extended Data Fig. 1b). This apparent paradox results from H₂O condensation in the hydrothermal system, which is the same process heating the rocks...*”

Fig. 3 Explain what sense in the scale change between modeled and CO/CO₂ temperatures

We have explained this difference in the caption of the figure 5: “...*The modeled temperatures, which refer to the central deeper zone of the computational domain (“Temperature box” in Fig. 4), have the same temporal evolution but are systematically higher than the CO-CO₂ temperatures, which reflect the thermal state of the upper part of the hydrothermal system ...*”

The uplift for Yellowstone is rather linear in contrast to exponential or power-law for other volcanoes.

As highlighted in the inset of Fig. 7a, the first period of the Yellowstone uplift was characterized by an accelerating trend. The figure below clearly shows that initially (for 1.0–1.5 years) the Yellowstone uplift followed an accelerating trend, after which the deformation slowed. This process is described in the text as follows: “...such as in the case of the Yellowstone caldera, where an initial exponential/power-law acceleration of ground uplift during 2004–2008 was followed by a deceleration...”

Reviewer #3

This manuscript proposes the idea that distinct solubilities of H₂O and CO₂, the two primary volatile species in magmas, lead to distinct open-system volatile release as a function of pressure, and that the transition to dominant H₂O exsolution during gradual decompression may control volcanic unrest and eruption. This work builds on previous studies by these authors on deformation and hydrothermal phenomena associated with recent unrest at Campi Flegrei, although there is discussion of other volcanoes. There are some interesting ideas in here that are worthy of publication in Nature. My main comments as detailed below relate to whether the connection of ideas to data is compelling. I will also note that there are slight English language issues that need to be addressed.

Thank you for the positive comments. We can confirm that the manuscript has been edited thoroughly by a native English speaker working for a professional English editing service.

The concept of a critical degassing pressure makes sense. It would seem that more justification of the open system assumption is warranted. You need to get bubbles out of the magma chamber, and this introduces a timescale that must be met (bubble rise, and creep closure in mushy wall rocks) that may depend on magma composition, thermal state and depth.

We agree with this comment, and have decided to restructure that part of the manuscript to provide more justification of the open-system assumption. The formation, growth, and leakage of a bubble foam layer occurs over timescales of months (up to 1 year) in low-viscosity mafic melts (see Menand and Phillips, 2007, Paonita et al. Geology, in press), which is much faster than the decennial trends in the He-N₂-CO₂ composition at CFC. The same order of timescale can be estimated for gas release through the roof of a reservoir (Paonita et al., 2016). This result can be confidently extended to the case of CFC. The new text discussing our open-system assumption is as follows: *"Our calculations were performed under open-system conditions since the long-lasting variations in the fumarole-gas composition at CFC cannot be reproduced in a closed system, instead requiring efficient separation of gas from the magma²¹. The large amount of magmatic fluids released by CFC manifestations⁴⁰ also supports an ongoing open (rather than closed) magma-degassing behavior. Our open-system Rayleigh-type degassing model assumes that volatiles are continuously separated from magma at each decompression step. However, the release of magmatic fluid from CFC surface manifestations actually shows a pulsed (noncontinuous) behavior (Extended Data Fig. 1 and 2), which suggests a mechanism in which periods of closed-system ascent alternate with episodes of system opening and gas release (i.e., a multistep degassing process). Tests show that such multistep degassing can be adequately reproduced as an open-system degassing process provided that there are numerous and recurrent system-opening events, as is likely to be the case. Open-system degassing is not only observed at CFC^{41, 42}. We argue that such degassing behavior ultimately can result from the complex geometry of crustal volcano plumbing systems, whose intricate networks of fractures, dikes, sills, and small reservoirs⁴³⁻⁴⁶ facilitate the segregation of gas from melt, and the loss of volatiles from a foam layer⁴⁷. Foam growth in low-viscosity mafic melts takes place over timescales of months to a few years^{47, 48}, which is faster than the observed decennial trends in gas composition. While we therefore favor an open-system scenario, we also show examples of model degassing simulations in closed-system conditions (Fig. 2b) to demonstrate that a CDP can be reached even in that type of system, despite the mass of released volatiles varying less markedly than in open-system conditions. We conclude that the concept of the CDP applies over a wide range of magmatic conditions."*

Doesn't the release of magmatic gases also result in precipitation of hydrous minerals around the magmatic system? Does this affect the mass you predict? What amount of heat would this involve.

Below we distinguish two classes of process in which hydrous minerals may take a role (as described below): (i) crystallization in the magmatic environment and (ii) hydrothermal minerals.

1. Studies at CFC indicate that hydrous minerals do not crystallize from trachybasalt melts, but only in the later stages of magma evolution (Cannatelli 2012, Stock et al., 2015). Apatites would contain only low levels of water even in evolved melts (Stock et al., 2015). Magma in crystallized hydrous phases is therefore unlikely to affect our results. Mafic magmas do not exsolve saline brines during their decompression (Carroll 2005 and references).

2. Water-bearing minerals could form by rock alteration as magmatic fluids migrate from magma to the hydrothermal system. These phases have been detected in geothermal wells drilled at CFC during the 1980s. Temperatures higher than 400°C have been found in the deeper wells (i.e., San Vito wells), along with hydrothermal to thermometamorphic mineral paragenesis (including hydrous minerals such as micas, amphiboles, and epidotes). Our model does not consider these reactions because the timescale of formation and volume of these newly formed minerals are unconstrained. In the revised version of the manuscript we explicitly state the limits of our modeling regarding the occurrence of secondary processes: *“...The composition (CO₂/H₂O ratio) of the injected magmatic gas phase is based on results of our magma-degassing models (see Methods and Fig. 1). We highlight that these model magmatic CO₂/H₂O ratios can only approximate the composition of fluids entering the real hydrothermal system, since the model does not account for secondary processes potentially occurring along the magma-to-hydrothermal gas cooling path...”* However, in the Discussion we observe the following: *“...The overall good match between the model results and observations (Figs. 5 and 6) supports the validity of the model setup and underlying assumptions. Although we can only guess the real CO₂/H₂O ratio signature of the feeding magmatic gas phase, there is a good fit between the modeled and observed trends if this ratio decreases over time. This represents compelling evidence for a decompressing magma trigger of the accelerating deformation at CFC, and implies that secondary processes in the magma-to-hydrothermal transition zone are likely to exert only marginal effects...”* The marginal role of secondary processes is probably justified by the high magmatic gas flux, and by the slowness of mineral precipitation compared to the relatively rapid fluid transfer from the magmatic zone to the hydrothermal system.

Regarding the application to Campi Flegrei. It would seem that some assessment of model uncertainty is required. Do the results change if permeability values or ratio of horizontal/vertical change? I understand that these are tuned to match the observed lag between ground deformation and fumarole emission, but the delay should depend on what the source of deformation is (expansion due to heating in the subsurface or magmatic source) and what the rheology is (viscoelastic/plastic effects such as are discussed here would modify the lag). I expect that such considerations are important for the total amount of gas released as well. We agree with the reviewer that the rock rheology in general, and permeability in particular, can significantly affect our results, as shown in previous studies (e.g., Todesco et al., 2003; Afanasiev et al., 2015). The physical properties of the rocks used in the model were derived from previous investigations at CFC. The permeability is that adopted in the reference case of Todesco et al. (2003) and in Chiodini et al. (2003). This permeability and porosity of the media are able to adequately reproduce some of the known features of the CFC hydrothermal system, such as the development under Solfatara of a gas zone that has been independently predicted by geochemical modeling (Todesco et al., 2003). Furthermore, when using this permeability it was possible to nicely replicate the shape and timing of the observed CO₂/H₂O peaks during episodes of magmatic gas injection (Chiodini et al., 2003). Chiodini (2009) subsequently noted for the first time a strong correlation between geochemical and geophysical signals (the lag between the two signals is an observation, and not a modeling result). Chiodini et al. (2012) accurately reproduced the observed lag between geochemical anomalies and the associated geophysical signals (earthquake swarms and pulsed deformations) in the physical model with the same permeability and porosity conditions that had been independently selected in previous studies (i.e., Todesco et al., 2003; Chiodini et al., 2003). These permeability and porosity conditions were not arbitrarily tuned to match the lag between geophysical and geochemical signals, which adds confidence to their realism. This result, and those reported in the manuscript (i.e., Figs. 5 and 6), suggest that the general behavior of the system is captured even with (i) the simplest assumption of constant permeability

and porosity conditions, and (ii) the limit of the model that works in the hydrothermal environment with water under subcritical conditions (i.e., without considering either viscoelastic or plastic effects).

Of course the relationship between pressure change and volume change depends on the fraction of bubbles in the chamber (e.g., Huppert and Woods 2002 JGR, also see recent work by Edmonds et al on different roles of H₂O vs CO₂). I see no discussion of this in relation to parameter estimation, but such would seem to be critical. Given the different heating capacities, it also would be good to place constraints on results due to CO₂/H₂O ratio uncertainties.

We understand the comment made by the reviewer here, and we do not rule out the presence of volume changes in the magmatic system. However, the inversion of geodetic data at CFC (Amoruso et al. 2013; D'Auria et al., 2010, 2015; Samsonov et al., 2015) indicates that the major source of the post 1983-1984 deformation was located at shallow depths (from 1 to 4 km, according to the different interpretations), which is above the inferred main magmatic system. Furthermore, we show an *"...overall good match between the model results and observations (Figs. 5 and 6)..."*, concluding that weakening of the rock due to alteration of its mechanical properties by heating contributes greatly to accelerating the deformation.

Moreover, we would like to stress again that the processes occurring in the magma chamber were not considered in our study, which instead focussed on the variation in the hydrothermal system caused by the arrival of magmatic fluids with different CO₂/H₂O ratios. In the revised version of the manuscript we clearly state in the introduction this main aim of our work (*"...The present study linked (i) magma degassing at depth with (ii) the resulting perturbation in the overlying hydrothermal system. Here we initially use the results of volatile saturation²⁵ models to demonstrate that decompressing magmas can reach a critical condition, which we refer as a critical degassing pressure (CDP), at which their ability to convectively transport heat is increased by a least an order of magnitude. We then use physical models²⁶ to show that magmatic volatiles released at the CDP, when injected into an overlying hydrothermal system, lead to extensive heating and expansion..."*)

What constrains the background temperature field? Is steady state a good assumption? It would also seem that the results will be highly sensitive to the background temperature field in the crust.

The steady-state assumption is derived from previous studies (i.e., Todesco et al., 2003; Chiodini et al., 2003) and is used only as in the initial condition of the simulation. These two previous studies showed that this assumption is reasonable because it accurately reproduces some of the main characteristics of Solfatara, including the presence of a shallow gas zone and the magnitude of the measured CO₂ flux. In any case, over the timescales we studied, the initial steady-state condition will mainly affect the absolute values of the simulated variables (e.g., temperature) and not their evolution over time, which is mainly controlled by the sequence of injection episodes (constrained by observed CO₂/CH₄ values), by their magnitude (constrained by observed CO₂/H₂O values), and by the H₂O-CO₂ composition of the injected fluids (constrained by the observed N₂/He values).

Finally, the exponential and power law fits to Rabaul, Sierra Negra, Campi Legrei and Yellowstone are very thought provoking. But it would be good to provide more details about how the deformation signals were compiled: is this vertical displacement averaged over some area? Presumably there is some uncertainty in the timing of first uplift? How does volatile composition/quantity affect this trend?

In the revised version of the manuscript we provide details on deformation data (i.e., that they are published vertical displacements). Data on the composition and flux of volatiles are available only at CFC, and we used these data to constrain the model. For instance, the different trends (e.g., of temperature and of the total injected fluids) obtained by using different volatile compositions are considered by comparing the results obtained in Simulation 1 (CO₂-rich gases with a constant composition) and Simulation 2 (in which gas composition becomes richer in H₂O over time), and this shows how the enrichment in H₂O causes heating of the system.

Reviewers' comments:

Reviewer #1 (Remarks to the Author):

I congratulate the authors on an excellent manuscript which is improved after changes made in review.

The principal results remain unchanged after review and, as I suggested, these are novel and exciting. The data and models presented are broad and well presented. The conclusions are well supported by the data and model predictions and are, therefore, robust. The writing is lucid and clear.

The extended methods section is now clear and concise and contains the information necessary to assess the application of Tough2 and the model that yields the CDP. The clarification of the power-law and exponential laws used to fit the acceleration of deformation are helpful.

I have a simple final question: If a packet of gas is exsolved from a liquid surface and then transported away, is it not then replaced by another packet of gas from below? This would result in a system that is not purely "open system" in the context of how I understand that Papale et al., (2006) is implemented to produce Figure 1. By this I mean that if we consider a reference point on an open and permeable gas network, then gas from below will be bypassing that point on the liquid-gas interface. My question is will this not buffer the saturation surface with the liquid that the authors compute and deviate from purely "open system" behaviour as they define it? The authors answer to this question is not something I consider an impediment to publication and just a question of interest.

Reviewer #2 (Remarks to the Author):

Review of the revised version of the manuscript #91384

The revised version looks much better and more to the point. The idea is clear and well supported by the modeling and a comparison of the modeled results and the observed data for Campi Flegrei. The only doubts occur when considering the published for CFC water isotope data. A good agreement between the modeled CO₂/H₂O increase in the volcanic vapors and the observed CO₂/H₂O increase in BG and BN fumaroles was explained as a loss of the water excess by condensation due to a specific thermal regime that has appeared when the magmatic CO₂/H₂O decreases. This means that the TOUGH modeling takes into account the whole water balance of the system and in this case the water isotopic variations could be estimated. The observed water isotopic composition is variable within a limited range, without any trends over time. And the releasing water vapor is closer to the meteoric than magmatic one. Why? A brief discussion, a few words, of this issue would be interesting. Another question is about changes in temperature. Why the Pisciarelli fumarole is changing its temperature but BG and BN fumaroles do not show any temperature trends? What does modeling says? Maybe I missed something in the text, but again, a sentence explaining this might be useful.

In my opinion, the manuscript is ready to be accepted after answering my questions.

Yuri Taran

Reviewer #3 (Remarks to the Author):

This manuscript is much improved, and I recommend publication after the following issues are addressed. I advocate playing down the application to Campi Flegrei and playing up the Critical Degassing Pressure idea. I don't think that you've learned anything robust about Campi Flegrei based on the work here.

Have alternative solubility models been used? It would seem that demonstrating the similarity claimed in the Results section is important.

The difference between your batch-type degassing predictions and real unsteady degassing measurements may also be explained by multiphase flow dynamics within the conduit, as well as complex conduit geometry. There is considerable literature on this, and its an interesting discussion point.

You suggest that a measure of potential energy release is marked by an inflection point in the gas solubility curve (the critical degassing pressure). But you do not provide a formula by which this is calculated. Are you computing the the second derivative of separated gas content with respect to pressure and finding where this equals zero (the inflection point)? Or what.

A "good match" has a precise definition in terms of minimization of residuals between model and observation. Has this been demonstrated?

I disagree that the good match between model and observations in fig 5 and 6 supports the validity of the model setup and underlying assumptions as stated in the discussion. You have not performed sensitivity tests to parameters, thus have not demonstrated that this is a unique solution or even that you are solving the correct equations. Please reformulate this statement and subsequent with something more qualified.

Along these lines - the predictive capability of this model for Campi Flegrei is not convincing. As pointed out in my first review, you dont know the intitial conditions or rheological parameters that have a large impact on results. So this is non-unique as far as I can tell. It seems slightly irresponsible to advocate increasing activity and potential hazard without doing robust sensitivity tests to parameters.

Leif Karlstrom

Reviewer #1 (Remarks to the Author):

I congratulate the authors on an excellent manuscript which is improved after changes made in review. The principal results remain unchanged after review and, as I suggested, these are novel and exciting. The data and models presented are broad and well presented. The conclusions are well supported by the data and model predictions and are, therefore, robust. The writing is lucid and clear.

The extended methods section is now clear and concise and contains the information necessary to assess the application of Tough2 and the model that yields the CDP. The clarification of the power-law and exponential laws used to fit the acceleration of deformation are helpful.

Authors: We wish to thank the reviewer for the positive evaluation.

I have a simple final question: If a packet of gas is exsolved from a liquid surface and then transported away, is it not then replaced by another packet of gas from below? This would result in a system that is not purely "open system" in the context of how I understand that Papale et al., (2006) is implemented to produce Figure 1. By this I mean that if we consider a reference point on an open and permeable gas network, then gas from below will be bypassing that point on the liquid-gas interface. My question is **will this not buffer the saturation surface** with the liquid that the authors compute and deviate from purely "open system" behaviour as they define it? The authors answer to this question is not something I consider an impediment to publication and just a question of interest.

Authors: we understand the reviewer's argument. However, the scenario invoked by the reviewer, e.g., that of vapour-buffered magmatic system, would not adequately reproduce the measured temporal trends in gas composition (i.e., Supplementary Fig. 4). The latter instead support degassing of a decompressing magma batch with a finite gas volume (non infinite, as in vapour-buffered case).

Reviewer #2 (Remarks to the Author):

The revised version looks much better and more to the point. The idea is clear and well supported by the modeling and a comparison of the modeled results and the observed data for Campi Flegrei.

Authors: We wish to thank the reviewer for the positive evaluation.

The only doubts occur when considering the published for CFC water isotope data. A good agreement between the modeled CO₂/H₂O increase in the volcanic vapors and the observed CO₂/H₂O increase in BG and BN fumaroles was explained as a loss of the water excess by condensation due to a specific thermal regime that has appeared when the magmatic CO₂/H₂O decreases. This means that the TOUGH modeling takes into account the whole water balance of the system and in this case the water isotopic variations could be estimated.

Authors: precise computation of water isotope variations during TOUGH modelling (as the reviewer suggests) is realistically not feasible. This would require: 1) knowledge of the isotopic signature for fluids initially present in the system, before the simulation run; 2) knowledge of isotopic compositions for pure fluid components involved in the simulation (i.e., the magmatic fluids injected at the bottom, and the shallow fluids, rain or locally seawater, whose fluxes into the system from the open top boundary is computed by the code); 3) computing the isotopic variations in each of the 850 cells of the computational domain, at each of the numerous time steps of the 30 years long simulation. This cell-by-cell processing should include, at each time step, the calculation of (1) the isotopic variations caused by fluid transfer from each given cell to its adjacent cells (in and/or out), (2) the isotopic fractionation during liquid-vapour phase changes, (3) the isotopic fractionation caused by oxygen exchange between H₂O and CO₂.

In summary, a new and complex code would be required that, starting from TOUGH2, calculate fluid isotopes step by step, concomitantly with the fluid-thermo-dynamic computations. This would be of great interest, but is realistically not possible in the frame of our work.

A simplified attempt to address the reviewer's comment is the back of the envelop calculation illustrated in Figure 1. In this example, we use the TOUGH2 simulated variations of the fumaroles X_{CO2} (gas checkpoint for gas composition in Fig. 4) to compute the expected variations of δ¹⁸O_{H2O} of the discharged steam (Fig. 1a, below) during the entire period of observation. To do this, we assume the simplified scenario of a fixed oxygen isotopic composition for the whole H₂O and CO₂ gas system (δ¹⁸O_{H2O+CO2} = +6.2‰ in the example of Fig. 1a) and the occurrence of CO₂-H₂O oxygen isotopic exchange at the discharge temperature of BG fumarole (~162°C). These TOUGH2 simulated variations of X_{CO2} and δ¹⁸O_{H2O} of the discharged steam are compared with measured compositions at BG fumarole (Fig. 1b). An overall agreement between model and observations can be deduced, that implies that oxygen isotope exchange does play a role at Campi Flegrei.

Fig. 1 Chronograms of (a) simulated and (b) measured fumarolic X_{CO2} and δ¹⁸O_{H2O}

The observed water isotopic composition is variable within a limited range, without any trends over time.

In order to answer this comment, we decided to publish the entire time-series of water stable isotopes for Solfatara fumaroles (see the new supplementary material). The reviewer's observation that isotopes do not show time related trends is biased by the fact that already published isotopic analyses are quite dated (published in Caliro et al. 2007). An updated dataset of water isotope compositions (Fig. 1a and 2) for the highest temperature fumarole (BG) of Campi Flegrei demonstrates, in fact, important temporal trends for both isotopes. Deuterium compositions, even if scattered, show a general increase of few $\delta\%$ units from 1983 to 2016 (Fig. 2, below).

Fig. 2 Chronogram of measured Deuterium concentration of fumarole BG

In contrast, ^{18}O data (Fig. 1b) show, after 2000, an evident time-related trend towards lighter compositions. This ^{18}O trend is anti-correlated to that of fumarolic X_{CO_2} (Fig. 1a, above). Oxygen isotopic exchange between H_2O and CO_2 molecules in the gas phase can well explain these anti-correlated variations of fumarolic CO_2 and ^{18}O . Fig. 1a compares the TOUGH2 simulated variations of X_{CO_2} and $\delta^{18}\text{O}_{\text{H}_2\text{O}}$ of discharged steam, calculated assuming $\text{CO}_2\text{-H}_2\text{O}$ oxygen isotopic exchange at discharge temperature. The simulated data show anti-correlated $\delta^{18}\text{O}_{\text{H}_2\text{O}}$ and X_{CO_2} variations that mimic those seen in the fumaroles (see Fig. 1 above). This agreement supports the occurrence of oxygen isotopic exchanges in fumarolic gases, a process that at Solfatara fumaroles was already proved based on analysis of oxygen isotopic composition of CO_2 (see position of the Campi Flegrei samples in Fig. 1 of Chiodini et al., 2000 which is reported here below).

Fig. 1. Relationship between the $1000 \ln \alpha_{\text{CO}_2\text{-H}_2\text{O}}$ in volcanic and hydrothermal fluids from Italian volcanoes and their emission temperature. Measured values range from -4.7 to $+37\%$ for discharge temperatures from 97° to 600°C . The polynomial best fit for the fumarolic data and the theoretical curve for equilibrium fractionation between CO_2 and H_2O in gas phase at the same temperatures (Bottinga, 1968; Richet et al., 1977) are reported for comparison. The good fitting of experimental results with theoretic data demonstrates the achievement of ^{18}O equilibrium in the volcanic gases within a wide temperature range.

.....and the releasing water vapor is closer to the meteoric than magmatic one. Why? A brief discussion, a few words, of this issue would be interesting.

Authors: The main question of the reviewer regards the presence (or not) of an isotopically recognisable magmatic component in Solfatara fumaroles. In our interpretation in the manuscript, we invoke steam condensation at depth as a driver for Campi Flegrei heating. As such, in order to verify the presence of a magmatic component, the effect of condensation has to be filtered out (e.g., the pre-condensation isotopic compositions have to be re-calculated).

We calculate the deuterium and oxygen isotopic composition of the pre-condensation vapour for the samples collected after 2000, i.e. in the period when steam condensation is thought to have occurred at depth (Chiodini et al., 2015). Calculations are initialised applying gas equilibria (H₂O-H₂-CO-CO₂ gas system) to measured BG fumarole compositions. This allows deriving equilibrium T-P conditions (T_{CO-CO_2} , X_{CO_2} , P_{H_2O}) and condensed steam fractions (f) in the vapour zone feeding Solfatara fumaroles (see Chiodini et al., 2015 for details).

The deuterium isotope compositions of hydrothermal steam ($\delta D_{H_2O,eq}$) are computed from the following equations of mass conservation and isotopic fractionation during condensation:

$$\delta D_{H_2O,eq} = f \times \delta D_{H_2O,l} + (1-f) \times \delta D_{H_2O,v} \quad (1)$$

$$1000 \ln \alpha_{l,v}(D) \sim \delta D_{H_2O,l} - \delta D_{H_2O,v} \quad (2)$$

In the equations, subscripts *eq*, *l* and *v* refer respectively to pre-condensation vapour (*eq*), condensed steam (*l*) and residual vapour (*v*). We assume condensation temperatures correspond to T_{CO-CO_2} .

We obtain pre-condensation δD values, ranging from -30‰ to -20‰ (Fig. 5a). The computed $\delta D_{H_2O,eq}$ values do not differ substantially from the measured $\delta D_{H_2O,v}$ because deuterium isotopic fractionations are low (close to 0) in the investigated temperature range (T_{CO-CO_2} from 210 to 240°C).

A similar, but more complex computational routine was used to derive information on oxygen isotopes. In fact, the oxygen isotopic composition of the H₂O+CO₂ gas system ($\delta^{18}O_{H_2O+CO_2}$) has to be considered. CO₂-H₂O oxygen isotopic exchange during gas cooling may dramatically change the isotopic composition of individual gas components (e.g., CO₂ or H₂O taken individually), while leaving the isotopic H₂O+CO₂ composition un-changed (Fig. 3, Chiodini et al., 2000). This effect, which is negligible in low CO₂ fumaroles, is of primary importance in high CO₂ fumaroles such as those of Solfatara.

Fig. 3 Variation of $\delta^{18}O_{H_2O}$ (H₂O line), $\delta^{18}O_{CO_2}$ (CO₂ line) and $\delta^{18}O_{H_2O+CO_2}$ (H₂O+ CO₂ line) caused by oxygen isotopic exchange during the cooling of a CO₂-H₂O gas mixture from magmatic to hydrothermal temperatures.

The following set of equations were used to derive $\delta^{18}\text{O}_{\text{CO}_2+\text{H}_2\text{O,eq}}$:

$$\delta^{18}\text{O}_{\text{H}_2\text{O,eq}} = f \times \delta^{18}\text{O}_{\text{H}_2\text{O,l}} + (1-f) \times \delta^{18}\text{O}_{\text{H}_2\text{O,v}} \quad (3)$$

$$1000 \ln \alpha_{l,v} (^{18}\text{O}) \sim \delta^{18}\text{O}_{\text{H}_2\text{O,l}} - \delta^{18}\text{O}_{\text{H}_2\text{O,v}} \quad (4)$$

$$\delta^{18}\text{O}_{\text{CO}_2+\text{H}_2\text{O}} = \chi_{\text{CO}_2} \times \delta^{18}\text{O}_{\text{CO}_2} + (1-\chi_{\text{CO}_2}) \times \delta^{18}\text{O}_{\text{H}_2\text{O}} \quad (5)$$

$$1000 \ln \alpha_{\text{CO}_2, \text{H}_2\text{O}} \sim \delta^{18}\text{O}_{\text{CO}_2} - \delta^{18}\text{O}_{\text{H}_2\text{O}} \quad (6)$$

The derived $\delta^{18}\text{O}_{\text{CO}_2+\text{H}_2\text{O,eq}}$ range from +6‰ to +7‰, and exhibit a shift toward more positive compositions since 2007-2009 (see Fig. 4).

Fig. 4. Variations of the computed $t\delta D_{\text{H}_2\text{O,eq}}$ (a) and $\delta^{18}\text{O}_{\text{CO}_2+\text{H}_2\text{O,eq}}$ (b) in the 2001-2016 period.

Finally, our estimated deuterium and oxygen isotopic compositions are plotted in the $\delta\text{D}-\delta^{18}\text{O}$ diagram (Fig. 5, below), where they also are compared with analytical data and isotope compositions of possible fluid sources at Campi Flegrei (i.e. meteoric water, seawater, magma).

We conclude from our calculations that the isotopic composition of Campi Flegrei fluids is consistent with the involvement of magmatic fluids. We however believe the above elaborations are too technical and out of the aims of this specific article, which primarily addresses the topic of differential release of H_2O and CO_2 from magma, and its implication in volcanic unrests. We also point out the involvement of magmatic fluids in Solfatara fumaroles, and the effects of $\text{H}_2\text{O}-\text{CO}_2$ oxygen exchange, have already been highlighted and discussed in previous work (Chiodini et al., 2000; Caliro et al., 2007). **While we prefer keeping most of the above material out the current manuscript, we have still added a new figure (Supplementary Fig. 6) and the following sentences to the text to addresses the reviewer's comment, and to confirm isotopes are accounted in the interpretation:** "Condensation of a mixed magmatic-meteoric vapor, followed by $\text{H}_2\text{O}-\text{CO}_2$ oxygen isotope exchange in the fumaroles' feeding conduits⁵¹, also well account for the observed hydrogen and oxygen isotope composition of fumarolic steam (Supplementary Fig. 6)".

Fig. 5 (Supplementary Figure 6). δD vs $\delta^{18}O$ diagram. Starting from the measured isotopes of fumarolic condensates (post-2000 samples of BG fumaroles, gray circles), the equilibrium $\delta^{18}O$ - δD composition of hydrothermal vapor (H_2O+CO_2 ; red circles) was calculated. Calculations were performed at reservoir temperature (T_c) and CO_2 molar fractions (X_{CO_2}), and considering the fractions of condensed steam (f) from reservoir to discharge. T_c , X_{CO_2} and f were estimated applying gas equilibria in the H_2O - H_2 - CO_2 - CO gas system, following the approach described in ref. 14. Computations involved solving a set of isotope mass balance and fractionation equations. Fractionation during water condensation and H_2O - CO_2 isotope oxygen exchange⁵¹ were taken into account. The re-computed $\delta^{18}O$ values refer to the whole CO_2+H_2O system. Based on these model-derived compositions, the isotope signature of CF steam samples is consistent with a mixed meteoric-magmatic origin undergoing condensation.

Why the Pisciarelli fumarole is changing its temperature but BG and BN fumaroles do not show any temperature trends? What does modeling says? Maybe I missed something in the text, but again, a sentence explaining this might be useful.

This observation is true, the largest temperature variation occurred at Pisciarelli fumarole. At BG, the hottest Campi Flegrei fumarole, temperature increased slightly, while temperature of fumarole BN decreased of a few °C (Fig. 6a).

Fig. 6 a) observed temperature variation at fumaroles BG, BN and Pisciarelli. b) Temperature variations simulated with TOUGH2 in the upper part of the axisymmetric domain (simulation 2 in the submitted ms). The isolines refer to the simulated volumetric gas fraction X_g

We believe the temperature decrease at BN may have been caused by a local process: shallow-level percolation of water-rich fluids into the BN fumarole channels. This process acts as to cool an originally BG-type gas (note that the two fumaroles are only 20m apart), and is supported by H₂O contents of BN fumarole being systematically (although slightly) higher than at BG. We thus consider BG as the most representative fumarole of the Solfatara crater.

The reason(s) for the different behaviour exhibited by Solfatara and Pisciarelli remain poorly known. Still, clustering of post-2000 Campi Flegrei seismicity in the Pisciarelli area (D'Auria et al., 2011) supports a **structural control on hydrothermal vapour influx**. Interestingly, **model simulations (Fig. 6b) also reproduce a larger temperature increase in the peripheral zones such as Pisciarelli, which is located off-axis to the central part of the gas plume**. We argue that this temperature increase may reflect accumulation of hot liquids (condensates) at the margins of the main vapour plume (underneath BG).

We added the following sentence in the caption of Fig.6: *“During the same time interval, temperature increased of only 3-4 °C at the highest temperature fumarole BG, implying clustering of hydrothermal influx on the eastern outer slope of Solfatara crater, where Pisciarelli is sited (Fig. 3).”*

Reviewer #3 (Remarks to the Author):

This manuscript is much improved, and I recommend publication after the following issues are addressed. I advocate playing down the application to Campi Flegrei and playing up the Critical Degassing Pressure idea. I don't think that you've learned anything robust about Campi Flegrei based on the work here.

Authors: thanks for the overall positive evaluation. We have accepted the reviewer's recommendation to tone down application to the Campi Flegrei unrest. This implied both a minor rearrangement of the main text and addition of specific comments (see below)

Have alternative solubility models been used? It would seem that demonstrating the similarity claimed in the Results section is important.

Authors: We added in the supplementary material a Figure that demonstrates the CDP is attained also using alternative solubility models, e.g., VolatileCalc (Newman and Lowenstern 2002). The following sentence was included in the revised text: "We stress that CDP conditions are obtained even using alternative solubility models, e.g. VolatileCalc³⁶" (Supplementary Fig. 3).

The difference between your batch-type degassing predictions and real unsteady degassing measurements may also be explained by **multiphase flow dynamics within the conduit**, as well as complex conduit geometry. There is considerable literature on this, and its an interesting discussion point.

Authors: We agree with the reviewer, and have incorporated a sentence on multiphase flow dynamics (and appropriate references) in the revised text: "...Open-system, unsteady degassing is not only observed at $C_{Fc}^{42, 43}$. During extrusive volcanic eruptions, pulsed degassing behavior can occur even at shorter timescales, and is thought to derive from multiphase flow dynamics within the conduit⁴⁴...".

You suggest that a measure of potential energy release is marked by an inflection point in the gas solubility curve (the critical degassing pressure). But you do not provide a formula by which this is calculated. Are you computing the second derivative of separated gas content with respect to pressure and finding where this equals zero (the inflection point)? Or what.

Authors: We clarified in the text the "formula" used to calculate the CDP. As the reviewer clearly states, we use the second derivative of separated gas content with respect to pressure. Note however that the inflection point refers to the first derivative ($\Delta\text{mol}/\Delta P$); therefore, the second derivative reaches a maximum at the inflection point (CDP), not a zero. We added this information in the text: "we set the CDP as the pressure value at which the second derivative of separated gas content with respect to pressure reaches its maximum".

A "good match" has a precise definition in terms of minimization of residuals between model and observation. Has this been demonstrated? I disagree that the good match between model and observations in fig 5 and 6 supports the validity of the model setup and underlying assumptions as stated in the discussion. You have not performed sensitivity tests to parameters, thus have not demonstrated that this is a unique solution or even that you are solving the correct equations. Please reformulate this statement and subsequent with something more qualified.

Authors: we have revised this statement as the reviewer requested. In the revised text, we have omitted reference to "good match" between model and observations. The entire discussion on comparison between model and observations, as well as implications for the CF unrest, has been revised. In the revised text, we also more clearly describe field of applicability of our model. The new text reads as: "We propose the CDP can help interpreting the current evolution of the C_{Fc} unrest. Escalation in hydrothermal activity, and increasing concentrations of fumarolic species more sensitive to temperature, point to an ongoing heating process of C_{Fc} . Our thermo-fluid-dynamic models here suggest that injection of magma-derived H_2O , becoming more voluminous and frequent in time, may well be controlling such heating. We caution

that, since our model does not account for rheological properties and heterogeneities of the system, it cannot be used to predict mechanical evolution, e.g., caldera deformation. We yet note a similarity between the temporal evolution of caldera uplift and our modeled hydrothermal temperature increase (Fig. 5c). We conclude, therefore, that magmas approaching the CDP may be contributing to accelerating caldera inflation, observed since 2005.”

Along these lines - the predictive capability of this model for Campi Flegrei is not convincing. As pointed out in my first review, you don't know the initial conditions or rheological parameters that have a large impact on results. So this is non-unique as far as I can tell. It seems slightly irresponsible to advocate increasing activity and potential hazard without doing robust sensitivity tests to parameters.

Authors: we have accepted the reviewer's invitation of toning down implications of our results for the CF unrest. We agree that insufficient constraints on initial conditions and on system's properties prevent us from indentifying our model as "unique" for Campi Flegrei. This is now fully stated in the revised manuscript: *"We caution that, since our model does not account for rheological properties and heterogeneities of the system, it cannot be used to predict mechanical evolution, e.g., caldera deformation."* We are thus well aware our study cannot attempt at fully replicating evolution of the real system, and that alternative models for the unrest cannot be excluded. However, we remind there is plenty of independent evidence at Campi Flegrei to suggest increasing activity and potential hazard (which does not require any sensitivity test but only good and numerous observations). Escalating activity at Campi Flegrei is a fact testified by accelerating deformation (totalling >20 cm in the last 2 years), seismicity (the last seismic swarm at Pisciarelli was registered the 29th of August this year), and augmented hydrothermal activity. In 2012, this increased activity induced the Commissione Grandi Rischi of the Italian National Civil Defence (DCP) to raise the alert level of the area to yellow (from the green base level). This increased activity was already subject of several scientific works (one which proposing intrusion of magma in the subsoil of Napoli at 3 km depth; D'Auria et al., 2015).

In this context, the novel aspect of our manuscript brings to light is that the "Critical Degassing" concept, and the magmatic-gas triggered heating it implies, is **a reasonable interpretative model** that explains the **temporal evolution** of the measured parameters at Campi Flegrei. In other words, we are not attempting to say our model is a single and unique solution, but that **its output is consistent with observations at the surface** (*"...We caution that, since our model does not account for rheological properties and heterogeneities of the system, it cannot be used to predict mechanical evolution, e.g., caldera deformation. We yet note a similarity between the temporal evolution of caldera uplift and our modeled hydrothermal temperature increase (Fig. 5c). We conclude, therefore, that magmas approaching the CDP may be contributing to accelerating caldera inflation, observed since 2005..."*).

Finally, given the sensitive (highly populated) area we are working on, we believe it would be irresponsible not to make our results publicly available.

REVIEWERS' COMMENTS:

Reviewer #2 (Remarks to the Author):

The authors have clarified all my doubts with an additional text and figures. For me, the manuscript is now ready for publication without more corrections, and I congratulate authors with their excellent and provoking work, that, I'm sure, will find a great interest among the volcanological community.
Yuri Taran

Reviewer #3 (Remarks to the Author):

I have read the review responses and new manuscript, and feel that the authors adequately addressed my concerns. I'd be happy to see this published.